# Self-Improving World Modelling with Latent Actions

## Abstract

Internal modelling of the world—predicting transitions between previous states $X$ and next states $Y$ under actions $Z$—is essential to reasoning and planning for LLMs and VLMs. Learning such models typically requires costly action-labelled trajectories. We propose SWIRL, a self-improvement framework that learns from state-only sequences by treating actions as a latent variable and alternating between Forward World Modelling (FWM) $P_\theta(Y|X, Z)$ and an Inverse Dynamics Modelling (IDM) $Q_\phi(Z|X, Y)$. SWIRL iterates two phases: (1) Variational Information Maximisation, which updates the FWM to generate next states that maximise conditional mutual information with latent actions given prior states, encouraging identifiable consistency; and (2) ELBO Maximisation, which updates the IDM to explain observed transitions, effectively performing coordinate ascent. Both models are trained with reinforcement learning (specifically, GRPO) with the opposite frozen model's log-probability as a reward signal. We provide theoretical learnability guarantees for both updates, and evaluate SWIRL on LLMs and VLMs across multiple environments: single-turn and multi-turn open-world visual dynamics and synthetic textual environments for physics, web, and tool calling. SWIRL achieves gains of 16% on AURORA-BENCH, 28% on ByteMorph, 16% on WORLD-PREDICTIONBENCH, and 14% on STABLETOOL-BENCH.[1]

## 1. Introduction

*Intrinsic world modelling* is a model's latent understanding of the environment and agent dynamics, i.e., trajectories of states and actions, which enables simulating possible futures without hallucinations. Large foundation models, such as Large Language Models (LLMs) and Vision-Language Models (VLMs), have arguably been shown to internalise world modelling to some extent during their training, thus enabling reasoning and planning without an external, specialised world model (Vafa et al., 2024; Qiu et al., 2024; Liu et al., 2025b; Chen et al., 2025b; Wang et al.; Xiong et al., 2026). For example, explicitly training models to predict the results of invoking tools (Guo et al., 2025b) or executing code (Copet et al., 2025) significantly enhances tool calling and coding tasks. This ability can also transfer to spatial reasoning tasks (Qiu et al., 2025; Tehenan et al., 2025) and reduce hallucinations that contradict external regularities (Liu et al., 2025a; Chen et al., 2025b).

While promising, the development of robust internal world models faces a significant bottleneck in data scalability. Current approaches rely heavily on execution logs or trajectories where observations are densely annotated with specific actions, because such annotations are naturally available in restricted environments (e.g., logs for tool calling or coding). However, for open-world tasks, collecting manual annotations for every transition is prohibitively expensive and intractable. Moreover, an additional challenge is the inherent ambiguity of *inverse dynamics*: a transition between two states may be explained by multiple valid actions, making purely supervised learning brittle when data is sparse.

Inspired by recent advances in self-improving learning for other applications (Wang et al., 2025a;c; Jin et al., 2025; Mao et al., 2025; Jin et al., 2025), we propose SWIRL (Self-improving World modelling with Iterative RL), a reciprocal optimisation framework to enhance the intrinsic world modelling of LLMs and VLMs from state-only sequences where the intermediate action is latent. We formalise world modelling as two components (Wang et al., 2025b; Qiu et al., 2025; Chen et al., 2025a): *Forward World Modelling* (FWM; predicting the next state $y$ given current state and latent action $x, z$), and *Inverse Dynamics Model* (IDM; inferring latent action $z$ given states $x, y$). In our reciprocal framework, we optimise the FWM $P_\theta$ to generate

---

[1]Anonymous Institution, Anonymous City, Anonymous Region, Anonymous Country. Correspondence to: Anonymous Author <anon.email@domain.com>.

Preliminary work. Under review by the International Conference on Machine Learning (ICML). Do not distribute.

[1]The code and models developed in this paper will be made available at [anonymised].

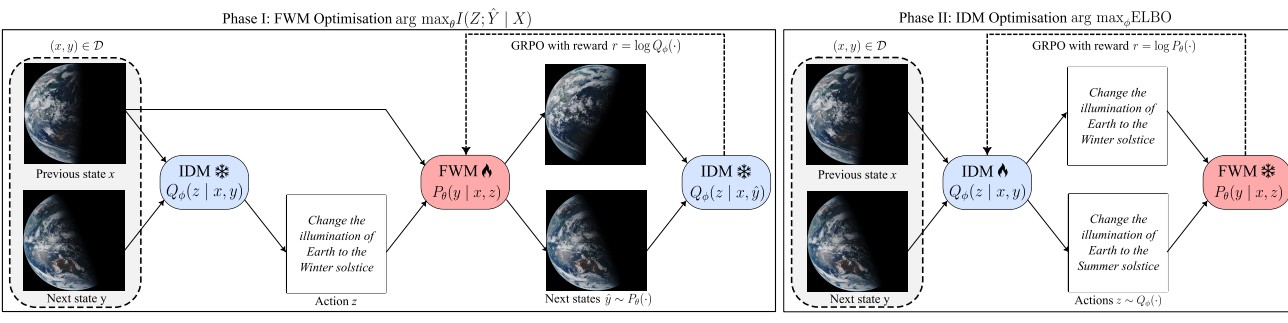

*Figure 1.* In SWIRL (Self-improving World modelling with Iterative RL), we facilitate the world modelling ability of foundation models (LLMs and VLMs) by modelling two components: Forward World Model (FWM) $P_\theta(y \mid x, z)$ and Inverse Dynamics Model (IDM) $Q_\phi(z \mid x, y)$. These components are iteratively optimised through RL (specifically, GRPO) in two distinct phases: I) the FDM acts as a policy and the IDM as a reward to ensure identifiability between actions and next states; II) the IDM acts as a policy and the FDM as a reward to ensure data fidelity to the state-only sequences. The KL term is omitted from the figure for simplicity.

futures that are consistently identifiable by the IDM, and the IDM $Q_\phi$ to infer actions that maximise the likelihood of the state dynamics predicted by the FWM. From a variational inference perspective, we theoretically prove that the optimisation of FWM is equivalent to maximising a lower bound on the Conditional Mutual Information $I(Z; \hat{Y}|X)$ (Barber & Agakov, 2003), and the optimisation of the IDM is equivalent to performing coordinate ascent on the Evidence Lower Bound (ELBO) of the log-likelihood $\log P_\theta(Y|X)$. Without relying on ground-truth action annotations, we optimise such a reciprocal framework with Group Relative Policy Optimisation (GRPO; Guo et al. 2025a). FWM and IDM take alternating roles of policy and reward, iterating until both components converge.

We validate our framework across three distinct environments on six benchmarks: open-world visual dynamics for VLMs on AURORA-BENCH, BYTEMORPH and WORLD-PREDICTIONBENCH, synthetic textual worlds on SCIENCE-WORLD, web HTML on MIND2WEB, and tool calling on STABLETOOLBENCH for LLMs. Empirical results confirm that our reciprocal self-improving allows models to learn effective dynamics from unlabelled state sequences, outperforming supervised fine-tuning baselines and achieving parity with larger, state-of-the-art models. Our contributions are summarised as follows:

- We propose a novel self-improving framework for world modelling in LLMs and VLMs, reciprocally reinforcing FWM and IDM without action annotations.

- We provide a rigorous theoretical proof that SWIRL corresponds to alternating between maximising Variational Mutual Information and Evidence Lower Bound.

- Empirical evidence on six benchmarks across visual, textual, web, tool calling environments demonstrates that SWIRL models effectively self-improve, leading to more enhanced forward world modelling capabilities.

## 2. Related Work

**Intrinsic World Models.** Recent research has explored whether internalised world models emerge in LLMs and VLMs through careful evaluation. Vafa et al. (2024) assessed whether the models' representations truly capture coherent world dynamics (Xiong et al., 2026). Similarly, Tehenan et al. (2025) and Qiu et al. (2024) found evidence that LLMs implicitly encode spatial and temporal relationships to some degree. World modelling also emerges naturally during the pre-training of large unified VLMs (Deng et al., 2025; Cui et al., 2025) and from video-based training (Chen et al., 2025b; Qiu et al., 2025).

In addition, previous work established that explicit world modelling can boost performance in downstream applications. For instance, Copet et al. (2025) proposed the Coding World Model for programming, and Lehrach et al. (2025) extended this approach to game-playing environments. Modelling the outcomes of function tool calls (Guo et al., 2025b) or planning (Li et al., 2025) had similar effects. As a result, dedicated post-training pipelines have been proposed to endow LLMs (Xie et al., 2025) and VLMs (Xiang et al., 2024) with explicit world modelling capabilities. Extensive benchmarks for evaluation have been proposed, including forward and inverse dynamics prediction (Chen et al., 2025a; Wang et al., 2025b; Gao et al., 2025).

**Self-Improving Learning.** Self-improving learning refers to models learning from their own (possibly curated) signals without external supervision. Huang et al. (2023) showed that LLMs can generate high-confidence answers and fine-tune themselves on these outputs to improve reasoning. Bensal et al. (2025) proposed a self-reflection and reinforcement learning loop where models analyse mistakes and retry, boosting performance on tasks like maths and function calling. Lee et al. introduce a curriculum in which models iteratively generate and filter correct answers, progressively tackling harder problems, while Zhao et al. (2024) demonstrate

that self-synthesised input–output pairs improve classification and generation quality. (Wang et al., 2025a) enhance self-improvement capabilities of agents with a skill library. LLM's coding and unit test generation capabilities can also co-evolve iterating on each other's outcomes (Wang et al., 2025c). Similarly, VLMs can refine visual and language reasoning using self-generated corrections (He et al., 2025).

In unified VLMs (Deng et al., 2025; Wu et al., 2024; Lin et al., 2025; Xiao et al., 2025), understanding performance often exceeds generation (Shi et al., 2025; Ma et al., 2025; Qu et al., 2025; Zheng et al., 2025; Zhang et al., 2025). A common strategy is then to use the understanding head as a critic to guide generation with carefully designed rubrics or heuristics (Mao et al., 2025; Jin et al., 2025; Qiu et al., 2026). Unlike these approaches, SWIRL theoretically and empirically proves the effectiveness of utilising the predicted likelihoods from understanding (action prediction) and generation (next state prediction) to establish a reciprocal cycle for VLMs where improvements in the generation head also enhance the understanding head, and vice versa.

## 3. Methodology

### 3.1. Task Formulation

We consider transitions from a source state $x \in \mathcal{S}$ to a target state $y \in \mathcal{S}$, mediated by a latent action $z \in \mathcal{A}$. Following Qiu et al. (2025) and Wang et al. (2025b), we parametrise world modelling using two components: i) *Forward World Modelling (FWM)*: $P_\theta(y|x, z)$, which predicts the next state given previous state and action; and ii) *Inverse-Dynamics Prediction (IDM)*: $Q_\phi(z|x, y)$, which infers the action given the state transition. The parameters $\theta$ and $\phi$ can be either disjoint or shared.

We consider four classes of environments with different observation–action formulations. In real-world visual environments, observations are pixel-level visual inputs and actions are specified in natural language. We study this setting using unified vision–language models (VLMs) capable of perceiving and generating interleaved image–text sequences. In synthetic textual environments, both observations and actions are expressed purely in language and are governed by an underlying simulator, which we model using large language models (LLMs). In web-based environments, states and actions correspond to raw HTML and interaction logs, respectively, while in tool-use settings, actions are tool calls and observations consist of the conversational context and tool execution outcomes.

### 3.2. SWIRL

**Intuition.** SWIRL alternates between two tightly coupled optimisation phases. In the first phase, we optimise the

---

**Algorithm 1** SWIRL

1: **Input:** Unlabelled dataset $\mathcal{D} = \{(x_i, y_i)\}$
2: **Initialise:** FWM $P_\theta$ , IDM $Q_\phi$
3: **Hyperparams.:** Group size $G$, Learning rates $\eta_\theta, \eta_\phi$
4: **repeat**
5:     === Phase I: optimise FWM ===
6:     **Freeze** IDM parameters $\phi$.
7:     **for** each batch $x \sim \mathcal{D}$ **do**
8:       Sample latent action $z \sim Q_\phi(z|x, y)$ for batch.
9:       Generate $G$ rollouts: $\{\hat{y}_1, \dots, \hat{y}_G\} \sim P_\theta(\cdot|x, z)$.
10:       **for** $k = 1$ to $G$ **do**
11:         Compute reciprocal reward via frozen IDM:
12:         $R_k \leftarrow \log Q_\phi(z|x, \hat{y}_k)$
13:       Compute Advantage $A_k$.
14:       Update $\theta$:
15:       $\theta \leftarrow \theta + \eta_\theta \nabla_\theta \big[\frac{1}{G} \sum_{k=1}^{G} A_k \log P_\theta(\hat{y}_k|x, z)\big]$
16:     === Phase II: optimise IDM ===
17:     **Freeze** FWM parameters $\theta$.
18:     **for** each batch $(x, y) \sim \mathcal{D}$ **do**
19:       Sample $G$ actions: $\{z_1, \dots, z_G\} \sim Q_\phi(\cdot|x, y)$.
20:       **for** $k = 1$ to $G$ **do**
21:         Compute reciprocal reward via frozen FWM:
22:         $R_k \leftarrow \log P_\theta(y|x, z_k)$
23:       Compute Advantage $A_k$.
24:       Update $\phi$ to maximise:
25:       $\phi \leftarrow \phi + \eta_\phi \nabla_\phi \big[\frac{1}{G} \sum_{k=1}^{G} A_k \log Q_\phi(z_k|x, y)\big]$ [†]
26: **until** Convergence or Max Iterations
27: **Return** optimised FWM $\theta^*$ and IDM $\phi^*$

[†]The KL term is omitted in the GRPO objective here for brevity's sake.

---

Forward World Model (FWM), $P_\theta$, to generate future states that are identifiable by the Inverse Dynamics Model (IDM). This enforces **identifiability** in forward prediction. In Eq. 2 below, we prove that this objective is equivalent to maximising a variational lower bound on the conditional mutual information $I(Z; \hat{Y}|X)$ (Barber & Agakov, 2003) , encouraging the predicted futures to retain maximal information about the underlying latent actions.

In the second phase, we optimise the IDM, $Q_\phi$, to improve **data fidelity** by inferring actions that best explain the observed state transitions under the learned FWM. We formally justify this step by proving that the resulting objective corresponds to maximising the Evidence Lower Bound (ELBO) of the inverse model (see Eq. 5 below).

**Objectives.** As depicted in Algorithm 1, we rely only on state-only sequences $(x_t, y_{t+1})$ as data. In the first phase of each iteration of SWIRL, we optimise FWM as a policy based on the frozen IDM rewards. Let $P_\theta$ denote the FWM policy parametrised by $\theta$. For each training step in GRPO (Guo et al., 2025a), we first sample a latent action $z_t \sim$

$Q_\phi(\cdot \mid x_t, y_t)$ from the IDM. Then, we sample a group of $G$ rollouts $\{\hat{y}_{t+1}^{(k)}\}_{k=1}^G$ from $P_\theta(\cdot \mid x_t, z_t)$. To evaluate their quality, we use a frozen IDM model $Q_\phi$ prompted for the IDM task, which estimates the likelihood of the action $z_t$ given the generated transition: $P_{\pi_\phi}(z_t \mid x_t, \hat{y}_{t+1}^{(k)})$. The reward for the $k$-th rollout is defined as

$$r_k = \log Q_\phi(z_t \mid x_t, \hat{y}_{t+1}^{(k)}).$$

This reward enforces that a predicted future is considered plausible only if IDM can retrospectively explain the action that led to the predicted state. We then optimise $P_\theta$ to maximise the advantage induced by these rewards.

After the FWM policy converges, we invert the optimisation direction to refine the IDM. In this phase, the optimised FWM model is frozen and used as the reward signal. The policy $Q_\phi$ is prompted for the IDM task, generating action candidates $\{\hat{z}_t^{(i)}\}_{i=1}^G$ given a state transition $(x_t, y_{t+1})$. Each candidate action is evaluated by querying the FWM for the likelihood of the target observation:

$$r_k = \log P_{\pi_\theta}(y_{t+1} \mid x_t, \hat{z}_t^{(k)}).$$

This reward favours data fidelity with respect to the state transition. We alternate between FWM and IDM optimisation phases, swapping the roles of policy and reward model until convergence. This reciprocal training paradigm ensures that improvements in forward prediction directly enhance action inference by the inverse dynamic model, and vice versa, driving the system toward a globally consistent both given all unlabelled observations.

### 3.3. Theoretical Analysis

Phase I encourages that generated futures are *distinguishable* (high mutual information), allowing the FWM to produce distinct outcomes for varied latent actions. Phase II encourages that inferred actions are *plausible* (high likelihood), allowing the IDM to map transitions to actions that the FWM can reproduce.

**Phase I: FWM optimisation via Variational Information Maximisation.** In this phase, we freeze $\phi$ and update $\theta$ to generate next states $\hat{y}$ that are identifiable by the inference model. We formalise this as maximising the Conditional Mutual Information (CMI) $I(Z; \hat{Y}|X)$ with respect to the *empirical belief distribution* of the model. Let $\tilde{P}(z|x) \triangleq \mathbb{E}_{y \sim \mathcal{D}(y|x)}[Q_\phi(z|x, y)]$ denote the marginal distribution of latent actions inferred by the IDM over the dataset.

**Theorem 3.1** (FWM Lower Bound). *Optimising the FWM to maximise the log-probability assigned to generated samples by the frozen IDM maximises a variational lower bound on the Conditional Mutual Information $I_{\tilde{P}}(Z; \hat{Y}|X)$ defined over the empirical belief distribution $\tilde{P}(z|x)$.*

*Proof.* The mutual information under the data-induced joint distribution $P(x)\tilde{P}(z|x)P_\theta(\hat{y}|x, z)$ is:

$$I(Z; \hat{Y}|X) = \mathbb{E}_{x \sim \mathcal{D}}\left[H_{\tilde{P}}(Z|X) - H(Z|\hat{Y}, X)\right]. \quad (1)$$

The entropy of the latent belief $H_{\tilde{P}}(Z|X)$ depends only on the frozen IDM and data distribution, and is thus constant w.r.t. $\theta$. Maximising CMI is therefore equivalent to minimizing the conditional entropy $H(Z|\hat{Y}, X)$. We use the variational posterior $Q_\phi(z|x, \hat{y})$ as a proxy for the intractable true posterior $P_\theta(z|\hat{y}, x)$ (Barber & Agakov, 2003) to bound the entropy term:

$$-H(Z|\hat{Y}, X) = \mathbb{E}_{x, z \sim \tilde{P}, \hat{y} \sim P_\theta}[\log P_\theta(z|\hat{y}, x)] \quad (2)$$
$$\geq \mathbb{E}_{x \sim \mathcal{D}}\mathbb{E}_{z \sim \tilde{P}(z|x)}\mathbb{E}_{\hat{y} \sim P_\theta(\cdot|x, z)}[\log Q_\phi(z|x, \hat{y})].$$

Substituting the definition of $\tilde{P}(z|x)$ creates the following objective:

$$\mathcal{J}(\theta) = \mathbb{E}_{(x,y) \sim \mathcal{D}}\mathbb{E}_{z \sim Q_\phi(\cdot|x,y)}\mathbb{E}_{\hat{y} \sim P_\theta(\cdot|x,z)}[\log Q_\phi(z|x, \hat{y})]. \quad (3)$$

This matches Algorithm 1: we sample trajectories $(x, y)$, infer $z$ using the IDM, rollout $\hat{y}$ using the FWM, and reward the FWM with the IDM's log likelihood. The gradient of Eq. 3 is $\nabla_\theta \mathcal{J}(\theta) = \mathbb{E}[\log Q_\phi(z \mid x, \hat{y})\nabla_\theta \log P_\theta(\hat{y} \mid x, z)]$, for which Group Relative Policy Optimisation (GRPO) offers an unbiased estimator:

$$\widehat{\nabla_\theta \mathcal{J}}_{\text{GRPO}}(\theta) = \frac{1}{G}\sum_{k=1}^G A_k \nabla_\theta \log P_\theta(\hat{y}_k \mid x, z), \quad (4)$$

where $A_k$ are the group-relative advantages derived from rewards $R_k = \log Q_\phi(z \mid x, \hat{y}_k)$. $\qquad\square$

**Phase II: IDM optimisation via ELBO Maximisation.** In the second phase, we freeze $\theta$ and optimise $\phi$. The goal is to infer actions $z$ that explain the transition $(x, y)$ in the FWM dynamics, while staying close to the valid prior. We treat the initialised model at each iteration as an *informative prior* $P(z|x) \triangleq \pi_{\text{ref}}(z|x)$. This corresponds to maximising a $\beta$-weighted Evidence Lower Bound ($\beta$-ELBO).

**Theorem 3.2** (IDM Lower Bound). *Optimising the IDM via GRPO with reward $R = \log P_\theta(y|x, z)$ and reference policy $\pi_{\text{ref}}$ maximises the $\beta$-ELBO objective where the prior is defined by $\pi_{\text{ref}}$.*

*Proof.* We seek to maximise the marginal log-likelihood $\log P_\theta(y|x)$. By introducing the variational distribution $Q_\phi(z|x, y)$ and the informative prior $P(z|x) = \pi_{\text{ref}}(z|x)$, we derive the $\beta$-ELBO:

$$\log P_\theta(y|x) = \log \mathbb{E}_{z \sim Q_\phi}\left[\frac{P_\theta(y|x, z)P(z|x)}{Q_\phi(z|x, y)}\right]$$
$$\geq \mathbb{E}_{z \sim Q_\phi(\cdot|x, y)}\left[\log P_\theta(y|x, z)\right]$$
$$- \beta D_{\text{KL}}(Q_\phi(z|x, y) \| P(z|x)) \triangleq \mathcal{L}_{\beta-\text{ELBO}}.$$

The GRPO objective is defined as the expected reward subject to a KL constraint against the reference policy:

$$\mathcal{J}_{\text{GRPO}}(\phi) = \mathbb{E}_{z \sim Q_\phi}[R(x, z, y)] \\ - \beta D_{\text{KL}}(Q_\phi(\cdot|x, y) || \pi_{\text{ref}}(\cdot|x)). \quad (5)$$

By setting the reward $R(x, z, y) = \log P_\theta(y|x, z)$ and identifying the prior $P(z|x)$ with the reference policy $\pi_{\text{ref}}(z|x)$, we observe that $\mathcal{J}_{\text{GRPO}}(\phi) \equiv \mathcal{L}_{\beta-\text{ELBO}}$. Thus, the GRPO update performs coordinate ascent on the ELBO. $\qquad\square$

## 4. Experiments and Results

### 4.1. Experimental Setup

**Models.** We choose Liquid (Wu et al., 2024) as our base VLM as the publicly best 7B-size unified VLM with an autoregressive architecture. The medium size limits the scale requirements for the compute infrastructure, and the autoregressive nature allows for applying GRPO off-the-shelf (without *ad-hoc* adaptations for diffusion-based generation). For text-based environments, we use Qwen-2.5-3B-Instruct (Qwen Team, 2024), a competitive mid-size LLM.

**SFT Warm-up.** Since our base VLMs/LLMs are general-purpose, they lack the specific interface capabilities required for environments (e.g., Liquid cannot natively predict image conditioning on both image and textual action). Prior to SWIRL, we conduct an initial SFT, which is strictly viewed as policy initialisation to ensure the model outputs valid actions and states; without this, a random policy would generate outputs rendering RL exploration impossible (as evidenced by Liquid's poor zero-shot performance in GEDIT-BENCH in Appendix C). For VLMs, we utilise image editing mixtures from PICO-BANANA-400K (Qian et al., 2025) and AURORA (Krojer et al., 2024).[2] For LLMs, we fine-tune on the half of environment-specific episodes, and remain the rest discarding the annotated actions for SWIRL.

**Iterative RL.** For SWIRL, we first conduct controlled experiments on the unlabelled video mixture from UCF-101 (Soomro et al., 2012), Movement-in-Times (Monfort et al., 2019), Kinetics700 (Kay et al., 2017), limiting training to the first phase IDM→FWM of SWIRL. This setup enables strict comparison with the bootstrapping baseline of Qiu et al. (2025) and allows us to observe stable convergence within a single epoch. After this controlled phase, we scale training by uniformly sampling 30K videos per iteration from a large-scale unlabelled video corpus, VIDGEN-1M (Tan et al., 2024). We extract the frame pairs from videos as in (Chen et al., 2025d). Each iteration is trained for one epoch, and the model alternates between FWM and IDM

---

[2]As a sanity check, we report the zero-shot and SFT performance of Liquid on GEDIT-BENCH compared to other VLMs in Appendix C.

optimisation phases as described in Section 3.2. This protocol ensures a clean comparison to baselines in (Qiu et al., 2025) while measuring the improvement across iterations.

**Benchmarks.** We evaluate visual world modelling under two settings. Firstly, for single-turn next-observation prediction, we use AURORA-BENCH and BYTEMORPH, both of which formulate dynamics prediction as action-conditioned image editing tasks with a strong emphasis on correctness in action-centric dynamics. Secondly, to evaluate long-horizon world modelling, we adopt WORLDPREDICTIONBENCH, which supports multi-step rollouts of up to four future observations across five subtasks, enabling a comprehensive assessment of FWM consistency.

To validate the generalisation of SWIRL, we conduct experiments across three grounded environments on LLM, each is a unique scheme of state transition. We utilise SCIENCE-WORLD (Wang et al., 2022) to evaluate physical dynamics, where LLM must predict the textual consequences of scientific actions within a simulated world. We also employ MIND2WEB (Deng et al., 2023), which challenges the model to forecast the updated HTML DOM tree following user interactions (e.g., clicks) on web elements. Finally, we assess functional tool dynamics via STABLETOOLBENCH (Guo et al., 2024), where the objective is to simulate the execution output of API calls conditioned on the current conversational state and the invoked tool.

**Baselines.** For visual world modelling, we compare SWIRL against a directly fine-tuned Liquid model (SFT). We reproduce two strong baselines from Qiu et al. (2025): TEST-TIME VERIFICATION, which selects the best sample from the fine-tuned Liquid based on IDM scores, and BOOT-STRAP, which fine-tunes Liquid on silver data annotated by the IDM. We also include specialised diffusion-based image editing models, including GoT (Fang et al., 2025), SmartEdit (Huang et al., 2024), InstructPix2Pix (Brooks et al., 2023), and the Chameleon model family, following the protocol of Qiu et al. (2025). To position our approach against recent (larger or diffusion-based) unified VLMs, we also evaluate BAGEL (Deng et al., 2025), OmniGen (Xiao et al., 2025), OmniGen2 (Wu et al., 2025), BLIP3o-NEXT (Chen et al., 2025c), and UniWorld-V1 (Lin et al., 2025).

For text-based environments, prior work under this formulation is limited. We therefore primarily compare against SFT baselines and larger LLMs to ensure a fair comparison.

**Evaluation Metrics.** For visual forward world modelling, we adopt GPT-4o-as-a-judge for holistic evaluation, following (Qiu et al., 2025; Fang et al., 2025) with a 10-point scheme. We also report DiscEdit (DE) and CLIP scores for AURORA-BENCH as in (Krojer et al., 2024). For text environments, we use BLEU, BERTScore, and ROUGE-L.

*Table 1.* **Quantitative comparison on AURORA-BENCH.** We report GPT-4o-as-a-judge scores, DiscEdit (DE), and CLIP scores across five datasets (and their Average). Blue cells indicate where SWIRL outperforms Liquid-SFT, its direct baseline.

| METHOD | MAGICBRUSH | | | ACTION GENOME | | | SOMETHING | | | WHATSUP | | | KUBRIC | | | AVERAGE | | |
|---|---|---|---|---|---|---|---|---|---|---|---|---|---|---|---|---|---|---|
| | GPT-4o | DE | CLIP | GPT-4o | DE | CLIP | GPT-4o | DE | CLIP | GPT-4o | DE | CLIP | GPT-4o | DE | CLIP | GPT-4o | DE | CLIP |
| UNIFIED VLMs | | | | | | | | | | | | | | | | | | |
| BAGEL-14B | 8.14 | 0.44 | 0.95 | 6.72 | 0.08 | 0.90 | 6.90 | 0.16 | 0.82 | 5.40 | 0.22 | 0.95 | 5.82 | 0.18 | 0.94 | 6.44 | 0.22 | 0.91 |
| OMNIGEN2 | 7.04 | 0.48 | 0.92 | 4.96 | 0.12 | 0.90 | 6.00 | 0.21 | 0.82 | 6.60 | 0.28 | 0.92 | 5.54 | 0.14 | 0.89 | 6.05 | 0.25 | 0.89 |
| BLIP3O-NEXT | 3.42 | 0.70 | 0.74 | 3.04 | 0.44 | 0.67 | 2.86 | 0.35 | 0.66 | 3.00 | 0.40 | 0.78 | 3.40 | 0.54 | 0.74 | 3.14 | 0.49 | 0.72 |
| OMNIGEN | 6.86 | 0.48 | 0.94 | 5.70 | 0.18 | 0.85 | 7.11 | 0.46 | 0.80 | 7.52 | 0.24 | 0.93 | 5.69 | 0.16 | 0.91 | 6.59 | 0.30 | 0.89 |
| UNIWORLD-V1 | 7.42 | 0.40 | 0.91 | 7.06 | 0.10 | 0.88 | 7.37 | 0.25 | 0.77 | 8.20 | 0.46 | 0.91 | 6.76 | 0.42 | 0.84 | 7.36 | 0.33 | 0.86 |
| EXISTING BASELINES | | | | | | | | | | | | | | | | | | |
| INSTRUCTPIX2PIX | 3.12 | — | — | 1.20 | — | — | 0.96 | — | — | 0.00 | — | — | 1.88 | — | — | 1.43 | — | — |
| GOT | 5.96 | — | — | 1.61 | — | — | 2.62 | — | — | 1.58 | — | — | 3.92 | — | — | 3.14 | — | — |
| SMARTEDIT | 6.71 | — | — | 3.08 | — | — | 2.81 | — | — | 0.76 | — | — | 3.70 | — | — | 3.41 | — | — |
| CHAMELEON-SFT | 2.52 | — | — | 2.48 | — | — | 3.11 | — | — | 0.88 | — | — | 7.30 | — | — | 3.26 | — | — |
| CHAMELEON-BOOTSTRAP | 3.27 | — | — | 2.74 | — | — | 3.11 | — | — | 0.98 | — | — | 7.30 | — | — | 3.48 | — | — |
| OUR BASELINES | | | | | | | | | | | | | | | | | | |
| LIQUID-SFT | 5.46 | 0.28 | 0.92 | 2.76 | 0.34 | 0.79 | 3.00 | 0.27 | 0.74 | 3.60 | 0.32 | 0.85 | 7.00 | 0.60 | 0.90 | 4.36 | 0.36 | 0.84 |
| LIQUID-BOOTSTRAP | 5.27 | 0.30 | 0.89 | 3.02 | 0.42 | 0.79 | 2.86 | 0.29 | 0.75 | 3.88 | 0.28 | 0.87 | 5.57 | 0.30 | 0.90 | 4.11 | 0.32 | 0.84 |
| LIQUID-SFT w/ TEST-TIME VERIFICATION | | | | | | | | | | | | | | | | | | |
| $N = 2$ | 5.71 | 0.14 | 0.93 | 3.00 | 0.38 | 0.80 | 3.40 | 0.21 | 0.76 | 4.38 | 0.36 | 0.86 | 6.18 | 0.44 | 0.90 | 4.53 | 0.31 | 0.85 |
| $N = 4$ | 6.10 | 0.14 | 0.93 | 3.38 | 0.36 | 0.81 | 3.44 | 0.25 | 0.76 | 4.38 | 0.40 | 0.87 | 6.04 | 0.38 | 0.90 | 4.66 | 0.31 | 0.85 |
| $N = 8$ | 6.18 | 0.16 | 0.92 | 3.28 | 0.36 | 0.81 | 3.74 | 0.21 | 0.74 | 4.48 | 0.28 | 0.86 | 6.28 | 0.46 | 0.90 | 4.77 | 0.29 | 0.85 |
| **SWIRL (IDM → FWM)** | 6.48 | 0.26 | 0.92 | 3.58 | 0.36 | 0.80 | 3.44 | 0.33 | 0.74 | 4.08 | 0.32 | 0.86 | 6.59 | 0.50 | 0.90 | 4.83 | 0.36 | 0.84 |
| **SWIRL (ITERATIVE)** | 6.62 | 0.30 | 0.92 | 3.52 | 0.48 | 0.80 | 3.96 | 0.21 | 0.74 | 4.32 | 0.36 | 0.86 | 6.96 | 0.54 | 0.90 | 5.06 | 0.38 | 0.84 |
| **SWIRL (ITER. + SHARE)** | 6.00 | 0.28 | 0.92 | 3.40 | 0.40 | 0.80 | 3.73 | 0.31 | 0.74 | 4.46 | 0.30 | 0.85 | 7.45 | 0.62 | 0.91 | 5.00 | 0.39 | 0.84 |

## 4.2. Single-turn Visual Dynamics Prediction

**AURORA-BENCH.** We present the quantitative evaluation of visual world modelling on AURORA-BENCH in Table 1. We compare SWIRL against state-of-the-art unified VLMs, specialised diffusion-based editing models, and ablations of our method. The primary comparison of interest is against Liquid-SFT, our direct supervised fine-tuning baseline. As highlighted in blue, SWIRL delivers consistent and significant improvements across all five benchmarks. Compared to the baselines from (Qiu et al., 2025), SWIRL surpasses the bootstrapping strategy using IDM to synthesise trajectories from unlabelled videos, and computation-heavy inference techniques like Test-time Verification (with up to $N = 8$ samples). Our best iterative strategy raises the average GPT-4o evaluation score from 4.36 (SFT) to 5.06, which also outperforms the non-iterative supervision from IDM only (SWIRL (IDM → FWM)) at 4.83, indicating the effectiveness of SWIRL by alternating policy and reward roles between FWM and IDM. Furthermore, despite starting from a weaker base model such as Liquid, and relying on a more lightweight post-training,[3] SWIRL remains highly competitive with other state-of-the-art unified VLMs such as OmniGen2 and BAGEL, while substantially outperforming diffusion-based editors like InstructPix2Pix. Qualitative examples for SWIRL are presented in Appendix J.

---

[3]We use only around 400K samples from (Qian et al., 2025; Krojer et al., 2024) to initialise the editing ability of Liquid.

*Table 2.* **GPT4o-as-a-judge on the ByteMorph Benchmark.** In addition to reporting unified VLMs' performance, blue cells indicate where SWIRL outperforms Liquid-SFT, its direct baseline.

| METHOD | Camera Zoom | Camera Motion | Object Motion | Human Motion | Inter-action | Average |
|---|---|---|---|---|---|---|
| BAGEL-14B | 32.89 | 49.34 | 49.33 | 45.03 | 40.89 | 44.48 |
| OMNIGEN | 47.76 | 52.17 | 44.88 | 43.59 | 40.92 | 44.40 |
| OMNIGEN2 | 49.08 | 55.39 | 61.67 | 65.52 | 61.63 | 60.14 |
| BLIP3O-NEXT | 24.21 | 13.16 | 24.33 | 34.22 | 28.53 | 27.28 |
| UNIWORLD-V1 | 55.53 | 68.03 | 55.91 | 48.10 | 58.03 | 55.56 |
| LIQUID-SFT | 57.23 | 56.51 | 43.13 | 38.07 | 40.70 | 43.38 |
| **SWIRL** | | | | | | |
| **(IDM→FWM)** | 57.37 | 50.13 | 62.50 | 48.69 | 56.53 | 54.57 |
| **(ITERATIVE)** | 54.08 | 58.22 | 58.69 | 48.08 | 54.50 | 53.77 |
| **(ITER. + SHARE)** | 53.16 | 55.86 | 62.10 | 53.78 | 53.81 | 55.72 |

**BYTEMORPH.** As the BYTEMORPH results in Table 2 show, all variants of SWIRL substantially raise the *Average* score over the Liquid-SFT baseline, e.g. SWIRL ITERATIVE goes from 43.38 to 53.77 (+26.4%). Performance on global camera control (*Camera Zoom/Motion*) remains comparable. This is expected since the unlabelled in-the-wild videos used during the RL phase (e.g., VIDGEN-1M) are predominantly static and provide limited supervision for camera control. In contrast, SWIRL yields pronounced improvements on *Object/Human Motion* and *Interaction*, indicating effective learning of fine-grained dynamics. Notably, SWIRL matches the performance of larger or more heavily supervised VLMs (e.g., BAGEL and UniWorld-v1).

*Table 3.* **Long-Horizon World Modelling on WORLDPREDICTIONBENCH.** We report GPT-4o-as-a-judge scores. The left section averages performance across the available horizon for each subtask (Turns 1–6 for EgoExo, EPIC, IKEA; Turns 1–4 for COIN, CrossTask). The right section details the decay of the *Overall* score at each prediction turn. Blue cells indicate where SWIRL surpasses Liquid-SFT .

| | AVERAGE TASK SCORE (HORIZON 1–6) | | | | | | OVERALL SCORE BY TURN | | | | | |
|---|---|---|---|---|---|---|---|---|---|---|---|---|
| **MODEL** | **COIN**[†] | **CROSS**[†] | **EGOEXO** | **EPIC** | **IKEA** | **AVG.** | **T=1** | **T=2** | **T=3** | **T=4** | **T=5** | **T=6** |
| BAGEL | 4.31 | 3.94 | 2.88 | 3.37 | 3.01 | 3.30 | 4.29 | 4.10 | 3.47 | 3.22 | 2.47 | 2.23 |
| OMNIGEN2 | 3.78 | 3.53 | 2.73 | 3.01 | 3.24 | 3.28 | 3.25 | 2.64 | 4.05 | 3.75 | 3.06 | 3.11 |
| BLIP3O-NEXT | 1.74 | 1.58 | 1.24 | 1.37 | 1.70 | 1.31 | 2.46 | 1.82 | 1.20 | 0.95 | 0.83 | 0.63 |
| OMNIGEN | 3.61 | 3.48 | 3.20 | 3.14 | 3.56 | 3.45 | 3.25 | 2.83 | 4.00 | 3.92 | 3.28 | 3.45 |
| UNIWORLD-V1 | 3.27 | 3.42 | 3.20 | 2.57 | 3.39 | 3.35 | 3.40 | 3.41 | 3.68 | 3.59 | 3.09 | 2.97 |
| LIQUID-SFT | 2.11 | 2.06 | 1.59 | 1.80 | 1.35 | 1.63 | 3.09 | 2.09 | 1.40 | 1.17 | 1.07 | 0.97 |
| **SWIRL (IDM → FWM)** | 2.61 | 2.45 | 1.84 | 1.95 | 1.56 | 1.89 | 3.24 | 2.42 | 1.85 | 1.59 | 1.25 | 1.08 |
| **SWIRL (ITERATIVE)** | 2.22 | 2.01 | 1.76 | 1.86 | 1.49 | 1.74 | 3.23 | 2.08 | 1.53 | 1.32 | 1.15 | 1.11 |
| **SWIRL (ITER.+SHARE)** | 2.06 | 2.28 | 1.82 | 1.66 | 1.43 | 1.68 | 2.95 | 2.04 | 1.53 | 1.24 | 1.23 | 1.11 |

*Table 4.* **Main Results on Textual Environments.** We evaluate performance on SCIENCEWORLD, MIND2WEB, and STABLETOOL-BENCH. We report BERTScore (BS) and ROUGE-L (R-L) for the first two tasks, and BLEU across all tasks for STABLETOOLBENCH as in (Guo et al., 2024). **Bold** indicates the best performance in each column, blue cells indicate where SWIRL outperforms Liquid-SFT .

| | SCIENCEWORLD | | MIND2WEB | | STABLETOOLBENCH | | | | | |
|---|---|---|---|---|---|---|---|---|---|---|
| **MODEL** | **BS** | **R-L** | **BS** | **R-L** | **ID HIGH** | **ID LOW** | **ID MED** | **OOD** | **OOD FAIL** | **AVERAGE** |
| QWEN-2.5-3B-INSTRUCT | 82.87 | 18.68 | 82.11 | 18.01 | 7.74 | 7.74 | 9.48 | 6.59 | 3.18 | 6.95 |
| QWEN-2.5-7B-INSTRUCT | 80.48 | 12.50 | 77.03 | 7.35 | 5.93 | 9.89 | 8.30 | 9.17 | 3.20 | 7.30 |
| QWEN-2.5-14B-INSTRUCT | 81.46 | 15.52 | 75.43 | 3.02 | 6.23 | 8.68 | 8.58 | 6.48 | 3.27 | 6.65 |
| QWEN-2.5-32B-INSTRUCT | 81.01 | 14.53 | 79.01 | 11.86 | 7.13 | 7.13 | 10.22 | 7.75 | 4.34 | 7.31 |
| OLMO-3-7B-INSTRUCT | 79.27 | 11.28 | 76.05 | 6.00 | 8.28 | 13.33 | 7.62 | 9.15 | **7.54** | 9.18 |
| DEEPSEEK-7B-CHAT | 81.90 | 16.35 | 80.18 | 14.32 | 5.38 | 15.17 | 4.44 | 8.52 | 4.17 | 7.54 |
| QWEN-2.5-3B-SFT | **96.06** | 89.73 | 92.37 | 48.97 | 16.03 | 12.87 | 17.51 | 14.86 | 2.99 | 12.85 |
| **QWEN-2.5-3B-SWIRL** | **96.06** | **89.92** | **92.44** | **49.10** | **16.47** | **16.90** | **21.20** | **15.57** | 2.92 | **14.61** |

## 4.3. Multi-turn Visual Dynamics Prediction

**WORLDPREDICTIONBENCH.** To assess *long-horizon* world modelling in addition to single-step prediction, we evaluate performance on WORLDPREDICTIONBENCH. The model must predict future observations autoregressively up to $T = 6$ time steps, using its own previous predictions as context. This setup tests the model's ability to maintain physical consistency and resist the compounding covariate shift inherent in sequential generation. Table 3 summarises the GPT-4o-as-a-judge scores across all subsets in WORLD-PREDICTIONBENCH.

The best variant of our models demonstrates a significant improvement in temporal consistency compared to Liquid-SFT. While SFT yields competitive performance at the immediate next step ($T = 1$), it suffers from rapid degradation as the horizon increases, dropping from a score of 3.09 to 0.97 by $T = 6$. In contrast, SWIRL (ITERATIVE) maintains significantly higher fidelity throughout the rollout trajectory, achieving a +14.4% relative improvement over the baseline at $T = 6$ (1.11). We also remark that repeated self-improvement is not beneficial in this dataset, as performance peaks at the first iteration (IDM → FWM).

## 4.4. Textual Environments

Table 4 shows the quantitative evaluation on textual environments. When applied to Qwen-2.5-3B-Instruct in physical and digital simulation environments (SCIENCEWORLD and MIND2WEB), both SWIRL and SFT are near-saturation in semantic accuracy (>92 BERTScore) and comparable in exact lexical matching (ROUGE-L). The advantage of our approach becomes most pronounced in STABLETOOLBENCH, which requires the simulation of API execution outcomes. Here, we observe substantial improvements in generalisation capabilities, with our method surpassing SFT by +4.03 BLEU on *ID-Low* and +3.69 BLEU on *ID-Medium* splits. This establishes a new state-of-the-art for this model scale, outperforming also significantly larger open-weight instruction models (e.g., Qwen-2.5-32B, DeepSeek-7B). Overall, this suggests that the SWIRL can generalise to internalise complex API dynamics effectively for LLMs.

## 4.5. Analysis

**Iterative RL.** We run an analysis to determine the potential for cumulative gain through iterative self-improvement beyond the first round. Figure 2 tracks the training dynam-

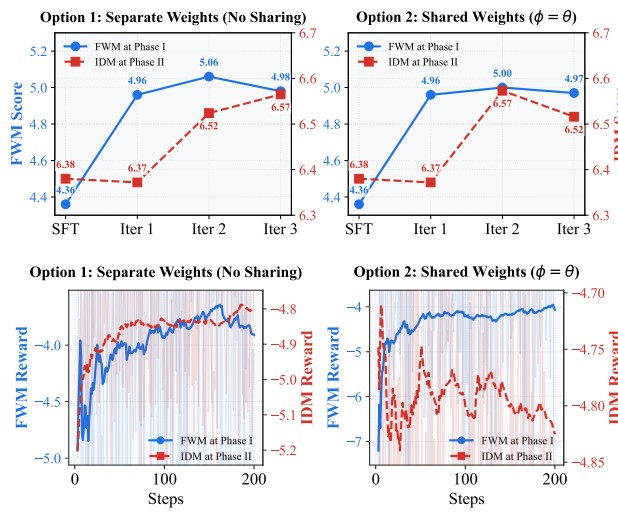

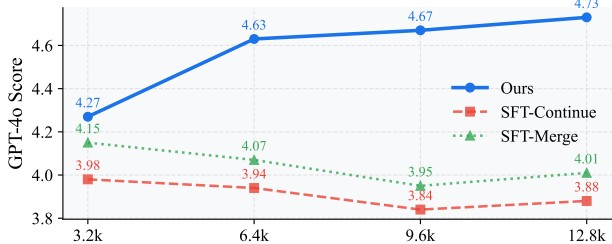

*Figure 2.* Performance of FWM and IDM in each iteration of SWIRL. We visualise the training dynamics across iterations for two settings: maintaining separate weights (Left) versus sharing parameters (Right). We present the evaluation performance on top, and the training rewards in the bottom. We present only the first iteration's reward curves for brevity.

ics across three iterations, where each iteration consists of updating the FWM using the current IDM as a reward (Phase I), followed by updating the IDM using the improved FWM as a reward (Phase II). We observe a clear *virtuous cycle*: enhancing the FWM's forecasting capability enables it to provide a more robust verification for action prediction, which in turn yields a more precise reward signal for the subsequent FWM updates. The reward curves demonstrate that FWM and IDM in a separate-weight setup improve the training rewards effectively, and their optimisation converges with GRPO.

**RL vs SFT.** To isolate the contribution of the optimisation objective, we compare SWIRL against direct SFT using the same set of unlabelled videos initially annotated by our IDM model. As shown in Figure 3 with details in F, SWIRL significantly outperforms the SFT baselines (both continued training and data merging), which stagnate or degrade as data scales. We attribute this disparity to the inherent ambiguity of visual dynamics and action verbalisation. SFT enforces a strict token-level imitation of the IDM's specific pseudo-labels; however, a single visual transition often corresponds to multiple valid descriptions. Forcing the model to mimic one specific verbalisation can lead to overfitting noise and suppressing valid alternative predictions. In contrast, our GRPO framework effectively relaxes this constraint by encouraging consistency assessed by IDM, rather than exact replication. By exploring the FWM's rollout space and action trajectories that the IDM model recognises as physically plausible, SWIRL learns a more robust and generalisable FWM that is not limited by

*Figure 3.* We compare our proposed SWIRL against SFT baselines (continual training and merging) across five benchmarks in AURORA-BENCH. The x-axis represents the number of training samples. Our method (solid blue line) demonstrates superior data efficiency, achieving higher GPT-4o evaluation scores.

the specific phrasing of the teacher annotations.

**Sharing $\theta$ and $\phi$.** We study the trade-off between parameter efficiency and training stability by comparing shared- and separate-weight designs (Figure 2 bottom and Table 1). Using *separate weights* yields stable optimisation, with the IDM score improving monotonically from 6.37 to 6.57 over three iterations (Figure 2, Left). In contrast, *shared weights* ($\theta = \phi$) reduce memory footprint but introduce instability, as reflected by a performance drop at Iteration 3 (to ~6.52).

This instability is also reflected in downstream performance: on AURORA-BENCH, shared weights slightly underperform the separate variant (5.06 vs. 5; Table 1), and on long-horizon evaluation (Table 3), the shared model consistently lags behind (e.g., 1.74 vs. 1.68 overall). We attribute this gap to gradient interference between the heterogeneous objectives of FWM (visual generation) and IDM (inferring the linguistic action), suggesting that more robust unification mechanisms are needed to effectively share representations across modalities (Shi et al., 2025; Ma et al., 2025; Qu et al., 2025; Zheng et al., 2025).

## 5. Conclusion

We introduce SWIRL, a unified framework for enabling VLMs and LLMs to intrinsically model future states conditioned on the current state and latent actions, without relying on human-annotated trajectories. By interpreting actions as latent variables and alternately optimising forward world modelling and inverse dynamics modelling objectives with GRPO, our method induces self-improving world modelling purely from unlabelled data. We provide theoretical guarantees establishing the learnability of each optimisation phase, formally linking our objectives to variational mutual information bounds and evidence lower bounds. Empirically, we demonstrate the effectiveness of SWIRL across diverse settings, including real-world visual dynamics with VLMs and textual environments (physical and digital simulations and tool calling) with LLMs, validating its generality.

## Impact Statement

This paper presents SWIRL, a framework that enables LLMs and VLMs to self-improve their internal world models using unlabelled data. By reducing reliance on expensive human-annotated trajectories, our work advances the efficiency and accessibility of training capable reasoning agents. This has positive implications for the development of general-purpose assistants that can better understand physical dynamics and digital environments.

However, we acknowledge potential societal consequences associated with these capabilities. First, our method utilises self-improving loops on unlabelled "in-the-wild" data. Without human curation, there is a risk that the model may internalise, reinforce, or amplify biases and harmful patterns present in the raw distribution of web and video data. Future work employing this framework should incorporate safety filters or constitutional AI principles within the reward mechanism to mitigate this risk.

Second, the improvements in Visual Dynamics Prediction and Web HTML interaction imply a step forward in generative video capabilities and autonomous web agents. While these advancements aid in creative tools and automation, they also lower the barrier for generating deepfakes or creating agents capable of bypassing web-based security measures (e.g., CAPTCHAs) or conducting automated interactions at scale. We emphasise the necessity of developing robust detection tools for synthetic media and implementing strict access controls and guardrails for autonomous agents operating in real-world web environments.

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

# A. Theoretical Derivation

In this section, we provide the detailed theoretical justification for SWIRL. We formalise the interaction between the Forward World Modelling (FWM) and Inverse Dynamics Model (IDM) as an alternating maximisation of *Identifiability* (via Variational Mutual Information) and *Data Fidelity* (via the Evidence Lower Bound).

We define the following notation:

- $x \in \mathcal{S}$: The source observation (current state $s_t$).
- $y \in \mathcal{S}$: The ground-truth target observation (next state $s_{t+1}$), distributed according to the data distribution $\mathcal{D}(y|x)$.
- $\hat{y} \in \mathcal{S}$: A generated target observation sampled from the forward model.
- $z \in \mathcal{A}$: The latent action driving the transition.

We employ two parametrised models:

1. **Forward World Modelling (FWM):** $P_\theta(\hat{y}|x, z)$, parametrised by $\theta$. FWM parametrised the environment's transition distribution.

2. **Inverse Dynamics Model (IDM):** $Q_\phi(z|x, y)$, parametrised by $\phi$. IDM parametrised an approximate posterior over latent actions given a state transition.

### A.1. Phase I: Learnability of the Forward Model (FWM)

In the first phase, we freeze the IDM parameters $\phi$ and optimise the FWM parameters $\theta$. Our objective is to generate trajectories $\hat{y}$ that are *identifiable* by the inference model. We formalise this as maximising the **Conditional Mutual Information** (CMI) between the latent action $Z$ and the generated observation $\hat{Y}$, conditioned on the source state $X$.

Crucially, in our algorithm, the latent actions $Z$ used for training are not sampled from a fixed uninformative prior, but are inferred from the ground-truth data using the current IDM. We denote this *Empirical Belief Distribution* as $\tilde{P}(z|x)$:

$$\tilde{P}(z|x) \triangleq \mathbb{E}_{y \sim \mathcal{D}(y|x)} \left[ Q_\phi(z|x, y) \right] \tag{6}$$

**Theorem A.1.** *Optimising the FWM to maximise the likelihood assigned by the frozen IDM to generated samples maximises a Variational Lower Bound of the Conditional Mutual Information $I_{\tilde{P}}(Z; \hat{Y}|X)$ defined over the empirical belief distribution.*

*Proof.* The Conditional Mutual Information under the joint distribution induced by the data and the frozen IDM is:

$$I_{\tilde{P}}(Z; \hat{Y}|X) = H_{\tilde{P}}(Z|X) - H(Z|\hat{Y}, X) \tag{7}$$

The term $H_{\tilde{P}}(Z|X)$ is the entropy of the marginal distribution of actions inferred from the dataset. Since $\phi$ is frozen in this phase and the dataset $\mathcal{D}$ is fixed, $\tilde{P}(z|x)$ is constant with respect to $\theta$. Therefore, maximising the mutual information is equivalent to minimizing the conditional entropy $H(Z|\hat{Y}, X)$.

The conditional entropy is defined as:

$$H(Z|\hat{Y}, X) = -\mathbb{E}_{x \sim \mathcal{D}} \mathbb{E}_{z \sim \tilde{P}(z|x)} \mathbb{E}_{\hat{y} \sim P_\theta(\cdot|x, z)} \left[ \log P(z|\hat{y}, x) \right] \tag{8}$$

The true posterior $P(z|\hat{y}, x)$ is intractable as it requires marginalising over the action space. We utilise the frozen IDM $Q_\phi(z|x, \hat{y})$ as a variational approximation. By the non-negativity of the KL divergence $D_{\mathrm{KL}}(P(z|\hat{y}, x)||Q_\phi(z|x, \hat{y})) \geq 0$, we have the lower bound:

$$\mathbb{E}_{\hat{y}}[\log P(z|\hat{y}, x)] \geq \mathbb{E}_{\hat{y}}[\log Q_\phi(z|x, \hat{y})] \tag{9}$$

Substituting this into the entropy term yields the variational lower bound for the CMI:

$$I_{\tilde{P}}(Z; \hat{Y}|X) \geq H_{\tilde{P}}(Z|X) + \mathbb{E}_{x \sim \mathcal{D}} \mathbb{E}_{z \sim \tilde{P}(z|x)} \mathbb{E}_{\hat{y} \sim P_\theta(\cdot|x, z)} \left[ \log Q_\phi(z|x, \hat{y}) \right] \tag{10}$$

Expanding the definition of $\tilde{P}(z|x)$ from Eq. (1), the optimization objective becomes:

$$\mathcal{J}_{\mathrm{FWM}}(\theta) = \mathbb{E}_{x \sim \mathcal{D}} \mathbb{E}_{y \sim \mathcal{D}(y|x)} \left[ \mathbb{E}_{z \sim Q_\phi(\cdot|x, y)} \left[ \mathbb{E}_{\hat{y} \sim P_\theta(\cdot|x, z)} \left[ \log Q_\phi(z|x, \hat{y}) \right] \right] \right] \tag{11}$$

$\square$

### A.2. Phase II: Learnability of the Inverse Model (IDM)

In the second phase, we freeze $\theta$ and optimise the IDM parameters $\phi$. We seek to maximise the log-likelihood of the observed ground-truth dynamics $\log P_\theta(y|x)$ (Data Fidelity). We model this via the Evidence Lower Bound (ELBO), treating the initialised policy at the start of the iteration as the reference prior, denoted $\pi_{\text{ref}}(z|x)$.

**Theorem A.2.** *Optimising the IDM via Group Relative Policy Optimisation (GRPO) with reward $R(x, z, y) = \log P_\theta(y|x, z)$ and reference policy $\pi_{\text{ref}}$ maximises the $\beta$-Evidence Lower Bound ($\beta$-ELBO).*

*Proof.* We express the marginal log-likelihood of the data by introducing the variational distribution $Q_\phi(z|x, y)$:

$$\log P_\theta(y|x) = \log \sum_{z \in \mathcal{A}} P_\theta(y|x, z)\pi_{\text{ref}}(z|x)$$

$$= \log \mathbb{E}_{z \sim Q_\phi(\cdot|x,y)} \left[ \frac{P_\theta(y|x, z)\pi_{\text{ref}}(z|x)}{Q_\phi(z|x, y)} \right] \tag{12}$$

Applying Jensen's inequality (concavity of log):

$$\log P_\theta(y|x) \geq \mathbb{E}_{z \sim Q_\phi} \left[ \log P_\theta(y|x, z) + \log \pi_{\text{ref}}(z|x) - \log Q_\phi(z|x, y) \right]$$

$$= \mathbb{E}_{z \sim Q_\phi} \left[ \log P_\theta(y|x, z) \right] - D_{\text{KL}}(Q_\phi(z|x, y) \,||\, \pi_{\text{ref}}(z|x)) \tag{13}$$

This is the standard Evidence Lower Bound. The Group Relative Policy Optimisation (GRPO) objective used in Algorithm 1 (Phase II) is defined as:

$$\mathcal{J}_{\text{GRPO}}(\phi) = \mathbb{E}_{z \sim Q_\phi(\cdot|x,y)} \left[ R(x, z, y) \right] - \beta \, D_{\text{KL}}(Q_\phi(\cdot|x, y) \,||\, \pi_{\text{ref}}(\cdot|x)) \tag{14}$$

By setting the reward to the FWM log-likelihood, $R = \log P_\theta(y|x, z)$, we observe that:

$$\mathcal{J}_{\text{GRPO}}(\phi) \equiv \beta\text{-ELBO}(\phi) \tag{15}$$

Thus, the IDM update step performs coordinate ascent on the $\beta$-weighted evidence lower bound of the data likelihood, ensuring the inferred actions explain the ground-truth transitions under the current forward dynamics. □

## B. Implementation Details

### B.1. Hyperparameters

For the LIQUID-SFT model, we fine-tune Liquid-7B (Wu et al., 2024) on PICO-BANANA-400K (Qian et al., 2025) and AURORA's training set. Training is performed for 5 epochs with a batch size of 128. We use a learning rate of $2 \times 10^{-5}$ with a cosine learning rate schedule, allocating 5% of the total training steps for warm-up.

For the non-iterative setup (SWIRL (IDM → FWM)), the model is trained with a batch size of 64 and a learning rate of $1 \times 10^{-6}$. A cosine learning rate schedule with 100 warm-up steps is applied. GRPO is used with a rollout size of 8 and a KL regularization coefficient $\beta = 0.1$.

For the iterative setup (SWIRL (ITERATIVE)), we use a batch size of 128 and a learning rate of $2 \times 10^{-7}$, together with a cosine schedule and 50 warm-up steps. In the SWIRL (ITERATIVE + SHARE) variant, the learning rate is increased to $5 \times 10^{-7}$ while all other settings remain unchanged. In both iterative GRPO setups, we set the decoding temperature to 0.75 and Top-$P$ to 0.96 to control the quality of predicted futures. Additionally, we apply a logit processor to constrain Liquid's generation to image tokens for FWM and textual tokens for IDM.

For LLM experiments on STABLETOOLBENCH, we fine-tune models using GRPO with DeepSpeed for memory-efficient distributed training. Optimization is performed with a learning rate of $5 \times 10^{-5}$ under a cosine learning rate schedule with 25 warm-up steps, using an effective batch size of 128. For each prompt, we sample 64 rollouts. The maximum prompt and completion lengths are set to 8,126 and 4,096 tokens, respectively. Unless otherwise specified, we use a KL regularization coefficient of $\beta = 0.1$. During GRPO rollout, we set the decoding temperature to 0.7 and Top-$P$ to 0.96. For MIND2WEB

and SCIENCEWORLD, we adopt the same configuration, except that the maximum prompt and completion lengths are both set to 4,096 tokens, and the number of GRPO rollouts is reduced to 16.

For VLM experiments, training is conducted using DeepSpeed on 32 NVIDIA H200 140GB GPUs for FWM and 8 NVIDIA H200 140GB GPUs for IDM. For LLM experiments, both FWM and IDM are trained on 8 NVIDIA Grace-Hopper (GH100) GPUs.

### B.2. Evaluation Prompt for GPT-4o as a Judge

For AURORA-BENCH, we use the same prompt as in Deng et al. (2025). For BYTEMORPH, we employ the official GPT-4o judge template provided by Chang et al. (2025). Since WORLDPREDICTIONBENCH (Chen et al., 2025a) does not natively support multi-turn image editing evaluation, we adopt the prompt template shown in Figure 4. This template is designed to balance the penalties for directly copying the source image and for excessive editing.

---

**WorldPrediction GPT-4o-as-a-Judge Prompt**

You are evaluating predicted observations for a procedural plan. The plan belongs to dataset {dataset}, sample {sample_uid}. Overall plan: {plan_description}. Focus on step {step_index} of {total_steps}, where the action is "{action_label}".

You are given the source observation (input to the model) and the candidate observation (model output). There is NO ground-truth reference image.

Your task is to judge whether the candidate correctly applies the stated action to the source. Do NOT score based on overall visual quality or realism.

Evaluation rules:

1. Infer what *must change* and what *must remain unchanged* given the action.

2. Compare candidate against source to check whether the required change is present.

3. If the candidate is identical or near-identical to the source when the action implies a visible change, assign a low score.

4. Penalise unintended changes outside the scope of the action.

Return a JSON object like `{"score":  <0-10>, "explanation":  "..."}` summarising correctness.

---

*Figure 4.* Prompt template used to instruct GPT-4o to evaluate multi-turn visual dynamics prediction in the WorldPrediction benchmark.

## C. Sanity Check Results for General Image Editing.

Table 5 reports a sanity-check evaluation of general image editing performance on GEDIT-BENCH. Although Liquid does not have native image editing support by design, we find that after our supervised fine-tuning (SFT) pipeline, using a mixture of AURORA (Krojer et al., 2024) and PICO-BANANA-400K (Qian et al., 2025), the resulting model (Liquid-SFT) acquires functional image editing behaviour. We illustrate the qualitative examples for general image editing in Figure 5.

Concretely, Liquid without SFT indicates a totally failure in GEDIT-BENCH, indicating the necessarily in the warm-up stage before SWIRL. Liquid-SFT achieves non-trivial scores in both semantic alignment and perceptual quality, confirming that the model can follow image editing instructions and produce coherent visual outputs. While Liquid-SFT remains significantly behind state-of-the-art proprietary and public image editing models, this result is not intended to be competitive; rather, it serves as a capability verification that our SFT pipeline successfully activates basic visual editing skills in a base model not originally designed for this task.

We emphasise that this capability plays an important role as a warm-up stage for subsequent world modelling objectives. In later stages, the model is trained to (i) predict the next visual observation given the current state (forward world modelling), and (ii) perform inverse dynamics modelling (IDM), where the model receives a pair of observations and predicts the intervening action in language. The ability to perform instruction-followed image transformations provides a minimal but necessary foundation for these more structured visual–temporal reasoning tasks.

Finally, the ablation study highlights the importance of PICO-BANANA-400K in our data mixture. Removing this component leads to a consistent degradation across all metrics (average score dropping from 3.06 to 2.43), indicating that Pico-banana

*Table 5.* General image editing performance on GEDIT-BENCH. As a sanity check, we evaluate the image editing performance of our SFT baseline (*Liquid-SFT*) against state-of-the-art image editing models. We report Semantics, Quality, and the Average score. Results indicate that our model maintains functional editing capabilities after our SFT pipeline.

| Models | Semantics | Quality | Average |
|---|---|---|---|
| **Proprietary** | | | |
| GPT-4o | 7.85 | **7.62** | **7.53** |
| Gemini 2.0 | 6.73 | 6.61 | 6.32 |
| **Public VLMs** | | | |
| Emu3.5-33B | **8.27** | 7.28 | 7.53 |
| BAGEL-14B | 7.36 | 6.83 | 6.52 |
| OmniGen2 | 6.35 | 6.03 | 5.57 |
| OmniGen-v1-diffusers | 5.51 | 5.37 | 4.62 |
| UniWorld-V1 | 5.05 | 6.85 | 4.80 |
| BLIP3o-NEXT-edit-VAE | 3.33 | 5.56 | 3.73 |
| Liquid Zero-shot | 0.04 | 1.43 | 0.03 |
| Liquid-SFT | 4.04 | 3.04 | 3.06 |
| Liquid-SFT *w/o* PICO-BANANA-400K | 2.92 | 2.87 | 2.43 |

*Table 6.* **AURORA-BENCH Evaluation of IDM.** We compare our proposed method against baselines on captioning/description quality. We report GPT-4 evaluation scores and reference-based metrics (BERTScore, ROUGE-L, BLEU). Blue cells indicate where ours positively improve Liquid-SFT.

| Method | GPT4o | BERT | R-L | BLEU |
|---|---|---|---|---|
| OmniGen2 | 6.15 | 14.5 | 15.3 | 7.5 |
| Chameleon | — | **45.0** | **44.0** | 20.0 |
| Liquid-AURORA | 4.52 | 31.0 | 31.0 | **28.0** |
| Liquid-SFT | 6.38 | 41.0 | 36.0 | 21.0 |
| **SWIRL (FWM $\rightarrow$ IDM)** | 6.38 | 40.1 | 39.2 | 20.3 |
| **SWIRL (Iterative)** | **6.58** | 40.3 | 40.3 | 21.1 |
| **SWIRL (Iter. + Share)** | 6.57 | 40.3 | 39.1 | 21.1 |

provides critical signal for stabilising and enriching visual instruction-following behaviour. This result suggests that even for non-native capabilities, carefully curated visual editing data is essential for preserving functional performance after SFT.

We include the qualitative examples produced by our approach for *general image editing* in Figure 5.

# D. IDM Analysis

**AURORA-BENCH Evaluation.** To validate the reliability of our reward mechanism, we evaluate the Inverse Dynamics Model (IDM) performance directly on AURORA-BENCH (Table 6). A fundamental premise of our framework is that inferring the action responsible for a state transition (IDM) is a more tractable task than generating high-dimensional future states (FWM). The results support this hypothesis: the IDM model achieves high action prediction accuracy (6.38 out of 10 as GPT score), serving as a strong discriminator. Additionally, we observe that the iterative paradigm can also leverage the reward signal from FWM, effectively improving GPT4o scores from 6.38 to 6.58 as our best approach. This stability is vital for our self-improving, ensuring that the reward signal remains informative and accurately penalises physical inconsistencies.

**Interpretability of Latent Actions and Reward Hacking.** A potential concern in learning with the latent actions is reward hacking, where actions might degenerate into short, distinctive ciphers (e.g., single keywords or artifacts) to be hacked for an easier optimisable FWM. We analyse the evolution of complexities of the generated actions across three

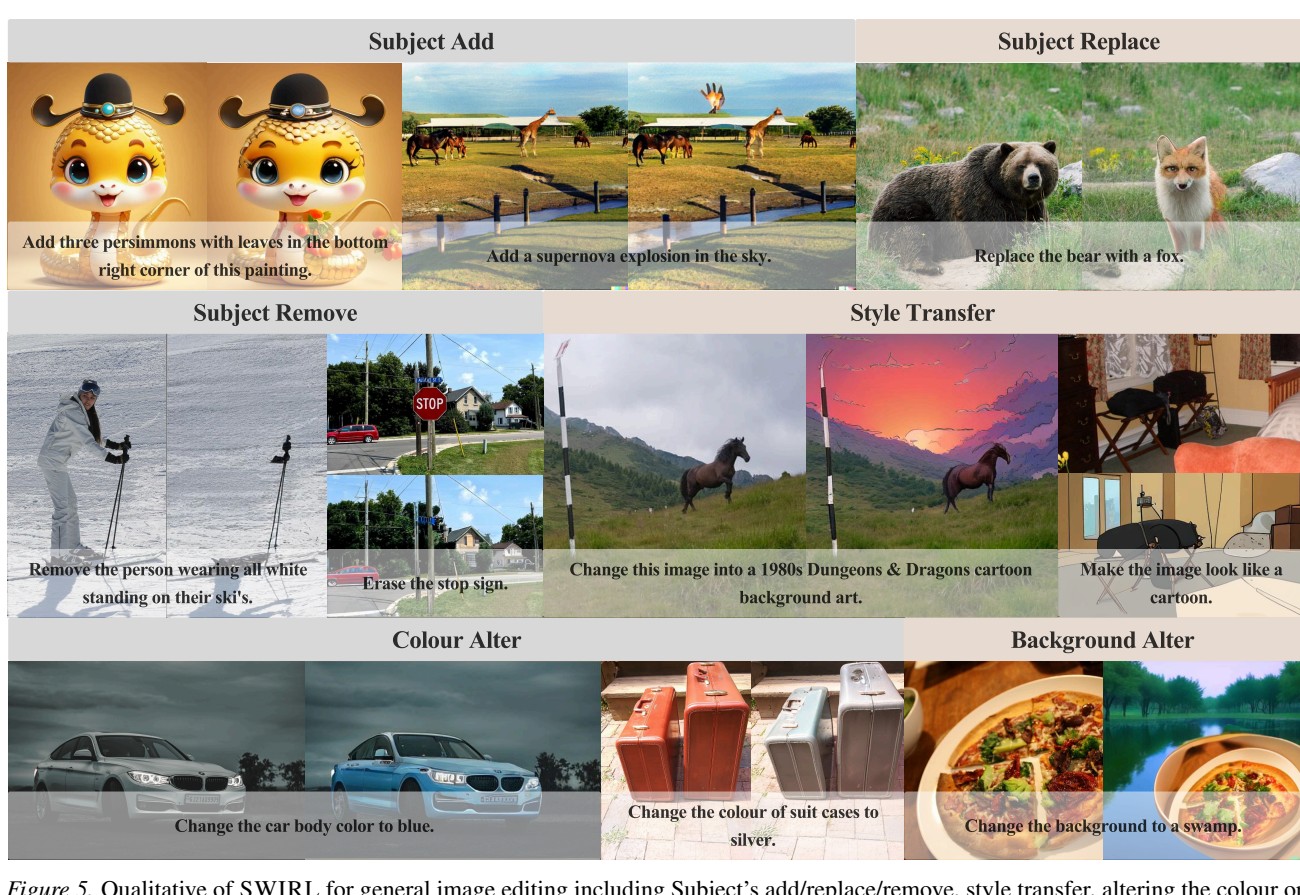

Figure 5. Qualitative of SWIRL for general image editing including Subject's add/replace/remove, style transfer, altering the colour or background.

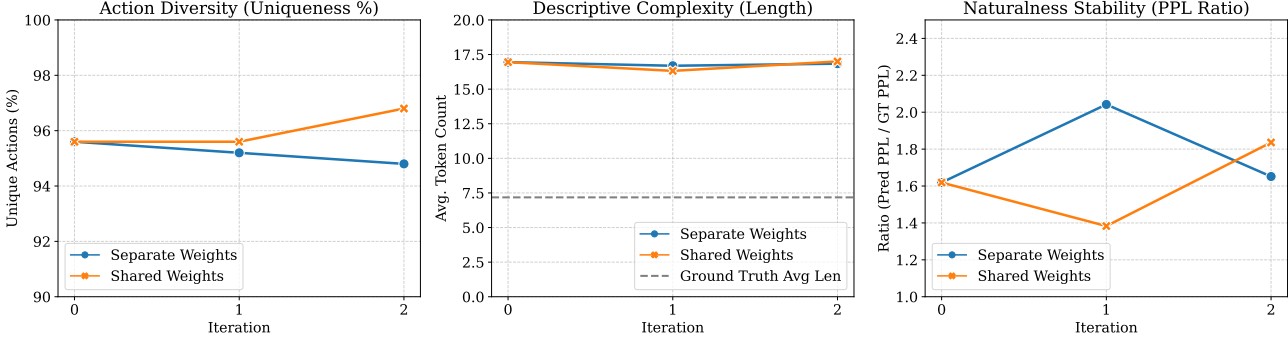

Figure 6. **Evolution of Latent Action Space throughout SWIRL.** We analyse the semantic properties of generated actions across iterations. *Uniqueness* indicates the percentage of distinct actions in the generated set. *Length* denotes average token count. *PPL Ratio* denotes the ratio of predicted perplexity from an LLM (specifically, GPT-2) on the predicted action against the ground-truth action, which measures the naturalness of the actions in language form. During SWIRL, the model maintains high diversity and naturalness without collapsing into short ciphers.

iterations for SWIRL in Figure 6.

Contrary to the hypothesis of reward collapse, we observe that the uniqueness of generated actions remains consistently high ($> 94\%$) across all iterations for both shared and separate weight configurations. The similar observation happens for naturalness that the predicted actions are quite stably natural (as evidenced by the predicted perplexity of an LLM, GPT-2) throughout SWIRL's training iterations, without collapsing into any unnatural artifacts that could be easily hacked by FWM or IDM. Looking into the lengths of predicted actions, the model does not "shortcut" by generating brief tokens as well; instead, the average action length (around 16.8 tokens) consistently exceeds the ground truth average (around 7.2 tokens). This indicates that the IDM learns to provide more descriptive and detailed instructions to ensure the transition is consistently identifiable, rather than drifting into a simplified cipher. Qualitative inspection confirms that predictions remain physically meaningful and there is no model collapse happening on SWIRL (e.g., *"tearing paper into two pieces", "Swap the positions of the two objects.", "turning a bottle upside down"*), preserving natural language interpretability.

## E. Inference-time Verification

As a sanity check for SWIRL, we investigate the validity of the *coverage hypothesis* intrinsic to our GRPO formulation. For the IDM reward to effectively guide learning, the set of sampled rollouts must contain at least one candidate that sufficiently approximates the ground-truth future state. We evaluate the "Best-of-$N$" performance of Liquid-SFT, where the candidate with the highest GPT4o score is selected, by scaling the number of rollouts $N$. In Table 1, we observe a monotonic improvement for INFERENCE-TIME VERIFICATION as $N$ increases, confirming that the base policy possesses the latent capability to generate high-fidelity transitions in multiple rollouts. This finding empirically validates our training premise: expanding the rollout space should increases the likelihood of capturing valid transitions, thereby providing the IDM discriminator with the necessary high-quality positive examples to reinforce.

## F. Detailed Results for Comparing SFT and SWIRL.

Table 7 presents a detailed comparison between SWIRL and two supervised fine-tuning baselines under controlled data budgets. Across all five benchmarks and all training sample counts, RL consistently exhibits superior data efficiency compared to both *SFT-Continue* and *SFT-Merge*. While SFT baselines often plateau or even degrade as more samples are introduced, RL shows a clear trend of performance improvement as additional data becomes available.

Notably, RL begins to outperform SFT at relatively small data scales (6.4K samples) and the performance gap widens as training proceeds. On Action Genome, Something, and Whatsup, Kubric, which emphasises structured physical dynamics, RL achieves the strongest or near-strongest performance at every scale, indicating that reward-guided optimisation better captures latent world structure than direct imitation.

Aggregated across all benchmarks, RL achieves the highest average score at every data scale, with gains becoming increasingly pronounced at larger budgets ($4.27 \rightarrow 4.73$). These results suggest that RL not only improves final performance but also utilises limited data more effectively, aligning with our hypothesis that IDM's rewarding mechanism provides a stronger inductive bias for intrinsic world modeling than purely supervised objectives.

## G. Detailed Results for WORLDPREDICTIONBENCH

Table 8 presents a granular analysis of predictive performance across six consecutive autoregressive turns on *WorldPredictionBench*. This evaluation creates a challenging stress test for temporal consistency, as errors generated in early turns compound over the predictive horizon.

We observe two primary trends. First, as expected, all models exhibit performance decay as the horizon increases ($N$ decreases as tasks are completed or fail). However, a clear divergence emerges between training paradigms. The direct supervised baseline (*Liquid-SFT*) suffers from rapid degradation, dropping from an overall score of 3.09 in Turn 1 to 1.17 by Turn 4. In contrast, our proposed SWIRL demonstrate significantly improved robustness against this compounding error. The *Ours (Best)* configuration, maintains a score of 1.59 at Turn 4. This indicates that the reciprocal cycle does not merely memorise single-step transitions but internalises more robust physical dynamics that persist over long-horizon rollouts. While large-scale unified models like *Bagel* provide a high upper bound, our method close the gap compared to SFT baseline.

*Table 7.* **Detailed Ablation: Data Efficiency (RL vs. SFT).** We report GPT-4o evaluation scores across five benchmarks and the aggregated average. We compare our proposed Reinforcement Learning (RL) fine-tuning against two Supervised Fine-Tuning (SFT) baselines: *SFT-Continue* (continual training with the additional samples) and *SFT-Merge* (we concatenate all samples and fine-tune model). Columns represent the number of training samples seen. **Bold** indicates the best performance for that specific sample count. Rows highlighted in blue denote our method.

| Benchmark | Method | Number of Training Samples | | | |
| | | 3.2k | 6.4k | 9.6k | 12.8k |
|---|---|---|---|---|---|
| MagicBrush | SFT-Continue | 5.50 | 4.94 | 4.70 | 4.56 |
| | SFT-Merge | **5.62** | 5.56 | 4.80 | 5.62 |
| | **SWIRL (RL)** | 4.71 | **5.56** | **6.61** | **6.19** |
| Action Genome | SFT-Continue | 2.62 | 2.72 | 2.82 | 2.88 |
| | SFT-Merge | **2.76** | 2.98 | 2.94 | 2.58 |
| | **SWIRL (RL)** | 2.60 | **3.32** | **3.67** | **3.38** |
| Something | SFT-Continue | 2.90 | 2.86 | 2.70 | 2.74 |
| | SFT-Merge | 2.92 | 2.54 | 2.66 | 2.56 |
| | **SWIRL (RL)** | **3.42** | **3.31** | **3.10** | **3.38** |
| Whatsup | SFT-Continue | 3.18 | 3.69 | 3.16 | 3.51 |
| | SFT-Merge | 2.69 | 2.62 | 2.74 | 2.86 |
| | **SWIRL (RL)** | **3.80** | **4.26** | **4.04** | **4.12** |
| Kubric | SFT-Continue | 5.81 | 5.51 | 5.90 | 5.79 |
| | SFT-Merge | 6.68 | 6.66 | **6.69** | 6.49 |
| | **SWIRL (RL)** | **6.90** | **6.70** | 6.51 | **6.65** |
| **Average (All)** | SFT-Continue | 3.98 | 3.94 | 3.84 | 3.88 |
| | SFT-Merge | 4.15 | 4.07 | 3.95 | 4.01 |
| | **SWIRL (RL)** | **4.27** | **4.63** | **4.67** | **4.73** |

*Table 8.* **Long-Horizon Evaluation on WorldPredictionBench.** We report GPT-4o evaluation scores across 6 consecutive prediction turns. We compare our proposed method variants against the direct SFT baseline (Liquid-SFT), the strong unified baseline (Bagel), and other state-of-the-art VLMs. Rows highlighted in blue denote our methods. The sample count ($N$) decreases in later turns as completed tasks are filtered out. **Bold** indicates the best performance among the Liquid-based family (SFT vs. Ours).

| Method | COIN | CrossTask | EgoExo | EPIC | IKEA | Overall |
|---|---|---|---|---|---|---|
| **Turn 1** ($N = 400$) | | | | | | |
| Bagel | 4.71 | 4.46 | 3.55 | 3.42 | 4.27 | 4.29 |
| Liquid-SFT | 3.37 | 3.41 | 2.75 | 3.20 | 2.57 | 3.09 |
| Ours (Iterative) | **3.60** | **3.44** | **3.11** | **3.30** | 2.48 | **3.23** |
| Ours (Iter.+Share) | 3.11 | 3.93 | 2.70 | 2.88 | 2.41 | 2.95 |
| Ours (Best) | 3.54 | 3.27 | 2.75 | 3.20 | **2.61** | 3.16 |
| *Other Baselines* | | | | | | |
| OmniGen2 | 3.44 | 3.59 | 2.26 | 2.73 | 2.67 | 3.05 |
| BLIP3o-NEXT | 2.65 | 2.76 | 2.15 | 2.33 | 2.20 | 2.46 |
| OmniGen-v1 | 3.41 | 3.46 | 2.64 | 4.35 | 2.73 | 3.25 |
| UniWorld-V1 | 3.26 | 3.59 | 3.23 | 2.83 | 3.89 | 3.40 |
| **Turn 2** ($N = 400$) | | | | | | |
| Bagel | 4.37 | 4.34 | 3.75 | 3.38 | 4.04 | 4.10 |

**Table 8 – continued from previous page**

| Method | COIN | CrossTask | EgoExo | EPIC | IKEA | Overall |
|---|---|---|---|---|---|---|
| Liquid-SFT | 2.42 | 2.37 | 1.83 | 2.08 | 1.52 | 2.09 |
| Ours (Iterative) | 2.25 | 1.90 | 2.06 | 2.25 | 1.78 | 2.08 |
| Ours (Iter.+Share) | 2.13 | 2.07 | **2.28** | 2.25 | 1.65 | 2.04 |
| Ours (Best) | **2.66** | **2.90** | 2.11 | **2.78** | **1.79** | **2.42** |
| *Other Baselines* | | | | | | |
| OmniGen2 | 3.13 | 2.80 | 2.09 | 2.38 | 2.08 | 2.64 |
| BLIP3o-NEXT | 2.05 | 1.80 | 1.58 | 1.65 | 1.60 | 1.82 |
| OmniGen-v1 | 2.99 | 3.22 | 2.34 | 3.55 | 2.37 | 2.83 |
| UniWorld-V1 | 3.25 | 3.49 | 3.06 | 3.38 | 3.86 | 3.41 |
| **Turn 3** ($N = 400$) | | | | | | |
| Bagel | 3.96 | 3.27 | 2.85 | 3.30 | 3.04 | 3.47 |
| Liquid-SFT | 1.46 | 1.27 | 1.42 | **1.75** | 1.17 | 1.40 |
| Ours (Iterative) | 1.61 | 1.39 | 1.62 | 1.65 | 1.33 | 1.53 |
| Ours (Iter.+Share) | 1.65 | 1.51 | 1.75 | 1.25 | 1.31 | 1.53 |
| Ours (Best) | **2.02** | **2.02** | **1.96** | 1.73 | **1.46** | **1.85** |
| *Other Baselines* | | | | | | |
| OmniGen2 | 4.16 | 4.34 | 3.30 | 4.00 | 4.14 | 4.05 |
| BLIP3o-NEXT | 1.25 | 1.05 | 1.06 | 1.25 | 1.24 | 1.20 |
| OmniGen-v1 | 4.05 | 3.95 | 3.81 | 2.93 | 4.48 | 4.00 |
| UniWorld-V1 | 3.57 | 3.90 | 3.25 | 2.93 | 4.35 | 3.68 |
| **Turn 4** ($N = 210$) | | | | | | |
| Bagel | 4.21 | 3.70 | 2.53 | 3.88 | 2.64 | 3.22 |
| Liquid-SFT | 1.20 | 1.20 | 1.29 | 1.32 | 1.04 | 1.17 |
| Ours (Iterative) | 1.40 | 1.30 | 1.38 | 1.41 | 1.23 | 1.32 |
| Ours (Iter.+Share) | 1.38 | **1.60** | 1.40 | 1.15 | 1.10 | 1.24 |
| Ours (Best) | **2.20** | **1.60** | **1.43** | **1.71** | **1.32** | **1.59** |
| *Other Baselines* | | | | | | |
| OmniGen2 | 4.40 | 3.40 | 3.29 | 3.18 | 3.95 | 3.75 |
| BLIP3o-NEXT | 1.00 | 0.70 | 1.00 | 1.06 | 0.89 | 0.95 |
| OmniGen-v1 | 4.00 | 3.30 | 3.79 | 3.21 | 4.31 | 3.92 |
| UniWorld-V1 | 3.00 | 2.70 | 3.83 | 2.44 | 4.32 | 3.59 |
| **Turn 5** ($N = 140$) | | | | | | |
| Bagel | — | — | 2.47 | 3.26 | 2.11 | 2.47 |
| Liquid-SFT | — | — | 1.23 | 1.30 | 0.91 | 1.07 |
| Ours (Iterative) | — | — | 1.26 | 1.17 | 1.09 | 1.15 |
| Ours (Iter.+Share) | — | — | **1.57** | **1.27** | 1.05 | 1.23 |
| Ours (Best) | — | — | 1.54 | 1.07 | **1.19** | **1.25** |
| *Other Baselines* | | | | | | |
| OmniGen2 | — | — | 2.91 | 2.83 | 3.21 | 3.06 |
| BLIP3o-NEXT | — | — | 0.91 | 1.03 | 0.71 | 0.83 |
| OmniGen-v1 | — | — | 3.57 | 2.57 | 3.43 | 3.28 |

**Table 8 – continued from previous page**

| Method | COIN | CrossTask | EgoExo | EPIC | IKEA | Overall |
|---|---|---|---|---|---|---|
| UniWorld-V1 | — | — | 3.29 | 2.10 | 3.40 | 3.09 |
| **Turn 6** ($N = 115$) | | | | | | |
| Bagel | — | — | 2.13 | 3.00 | 1.95 | 2.23 |
| Liquid-SFT | — | — | 1.04 | 1.17 | 0.88 | 0.97 |
| Ours (Iterative) | — | — | 1.12 | **1.39** | 1.02 | 1.11 |
| Ours (Iter.+Share) | — | — | 1.23 | 1.17 | **1.05** | **1.11** |
| Ours (Best) | — | — | **1.23** | 1.22 | 0.97 | 1.08 |
| *Other Baselines* | | | | | | |
| OmniGen2 | — | — | 2.54 | 2.96 | 3.39 | 3.11 |
| BLIP3o-NEXT | — | — | 0.73 | 0.91 | 3.55 | 0.63 |
| OmniGen-v1 | — | — | 3.04 | 2.22 | 4.05 | 3.45 |
| UniWorld-V1 | — | — | 2.58 | 1.74 | 0.50 | 2.97 |

# H. Detailed Results for Iterative Results

*Table 9.* **Detailed Iterative Dynamics (FWM & IDM).** We report the detailed breakdown of performance across three iterations for our two ablation settings: (1) Separate weights for Generator and Critic, and (2) Shared weights. Columns under "FWM Evaluation" represent the generation quality across five benchmarks. The "IDM Score" column represents the Critic's alignment accuracy. **Bold** numbers indicate the peak performance within each experimental setting.

| Setting | Iteration | FWM | | | | | | IDM |
|---|---|---|---|---|---|---|---|---|
| | | MagicBrush | AG | Something | Whatsup | Kubric | Avg | Avg Score |
| **Option 1: Separate Weights (No Sharing)** | | | | | | | | |
| | Iter 0 | 6.24 | 3.36 | 3.72 | 4.32 | **7.14** | 4.96 | 6.37 |
| | Iter 1 | **6.62** | **3.52** | **3.96** | **4.32** | 6.96 | **5.06** | 6.52 |
| | Iter 2 | 6.42 | 3.44 | 3.82 | 4.20 | 6.96 | 4.98 | **6.56** |
| **Option 2: Shared Weights** ($\phi = \theta$) | | | | | | | | |
| | Iter 0 | 6.24 | 3.36 | 3.72 | 4.32 | 7.14 | 4.96 | 6.37 |
| | Iter 1 | 6.00 | **3.38** | **3.73** | **4.46** | **7.45** | **5.00** | **6.57** |
| | Iter 2 | **6.38** | 3.18 | **3.73** | 4.32 | 7.27 | 4.97 | 6.52 |

Table 9 examines the iterative training dynamics of Forward World Modeling (FWM) and Inverse Dynamics Modeling (IDM) under two architectural choices: using separate weights for the FWM and IDM versus shared weights. We report results across three training iterations to highlight peak performance and learning progression.

The separate-weight configuration achieves the highest peak FWM performance, with Iteration 1 attaining the best average FWM score (5.06) and consistently strong results across all benchmarks. In particular, gains on MagicBrush, Action Genome, and Something indicate that decoupling the FWM and IDM allows each component to specialise more effectively, leading to higher-quality forward predictions when the system is optimally aligned.

While the shared-weight setup exhibits competitive and stable behaviour, it does not surpass the peak generative performance achieved by the separate-weight model. Notably, although IDM accuracy continues to improve slightly in later iterations for both settings, the separate-weight design reaches its optimal balance between FWM quality and IDM alignment earlier in training. This suggests that architectural decoupling enables more expressive forward modeling, even if later iterations yield diminishing returns.

Overall, these results indicate that separating FWM and IDM parameters is advantageous for maximising peak forward world modelling performance, motivating our choice of the separate-weight configuration in the main experiments where peak capability is the primary objective.

*Table 10.* **Ablation on GRPO Rollout Size** (*G*). We report the peak *Assemble* score achieved across all checkpoints for each rollout configuration. While larger group sizes ($G = 64$) achieve the best performance, moderate group sizes ($G = 16$) achieve comparable performance with much less compute.

| #Rollout | MB | AG | Something | Whatsup | Kubric | Avg. |
|---|---|---|---|---|---|---|
| $G = 8$ | 6.04 | 3.38 | 3.45 | 4.26 | 6.98 | 4.80 |
| $G = 16$ | 5.78 | 3.24 | 3.60 | 4.50 | 6.98 | 4.80 |
| $G = 32$ | 5.72 | 3.30 | 3.37 | 4.18 | 6.80 | 4.68 |
| $G = 64$ | 5.98 | 3.50 | 3.80 | 4.16 | 7.06 | 4.90 |

## I. Ablation on GRPO Rollout Size.

We analyse the effect of rollout size. The group size $G$ is a critical hyperparameter in GRPO, as it governs the accuracy of the baseline estimation and the diversity of the exploration within each optimisation step. We empirically evaluate $G \in \{8, 16, 32, 64\}$ across AURORA-BENCH, with results summarised in Table 10. We observe that scaling the rollout size generally improves model performance, achieving a peak average score of 4.90 at $G = 64$. We attribute this to the reduced variance in advantage estimation: a larger pool of generations provides a more robust approximation of the expected return, enabling the policy to distinguish high-quality trajectories more effectively. Notably, while $G = 64$ yields the best absolute performance—particularly in complex environments like Kubric, the smaller configuration of $G = 16$ remains highly competitive (4.80 avg.) while requiring much less memory and compute during the generation phase.

## J. Qualitative Examples

We provide some qualitative examples in Figure 7 to illustrate the visual predictions of SWIRL across three distinct benchmarks: AURORA-BENCH, BYTEMORPH, and WORLDPREDICTIONBENCH.

**AURORA-BENCH.** The top panel demonstrates results on the AURORA benchmark, covering various subsets such as MagicBrush, Something, Emu, Kubric, and Whatsup. These examples highlight the model's capability to perform diverse action-centric image editing tasks, ranging from the general editing (e.g., Add a `supernova explosion in the sky`) and background replacement (e.g., `changing to Yellowstone National Park`) to precise geometric transformations and spatial reasoning. For instance, in the Something and Kubric subsets, the model successfully executes actions concerning physical regularities like `flip the bottle upside down` and distinct spatial rearrangements like `moving a specific plate to the left`.

**BYTEMORPH.** The middle panel displays qualitative results for BYTEMORPH, focusing on fine-grained control over camera and object dynamics. We visualise three distinct categories: Camera Zoom, Camera Motion, and Object Motion. The results show the model's ability to synthesise coherent view changes, such as zooming out to reveal context (e.g., *the coastline and a second boat*) or shifting the camera angle upward. Additionally, the Object Motion example demonstrates the generation of localised movement, specifically raising an animal's head while maintaining scene consistency.

**WORLDPREDICTIONBENCH.** The bottom panel illustrates long-horizon predictions on the WORLDPREDICTIONBENCH. Here, we show multi-step predictions where the model generates subsequent states based on the previous predicted image as the context and textual action descriptions. The examples depict long-horizon consistency in procedural tasks, such as replacing a car key battery (Start → Put in battery → Close cover) and arranging bedding (Start → Take out cover → Arrange nicely).

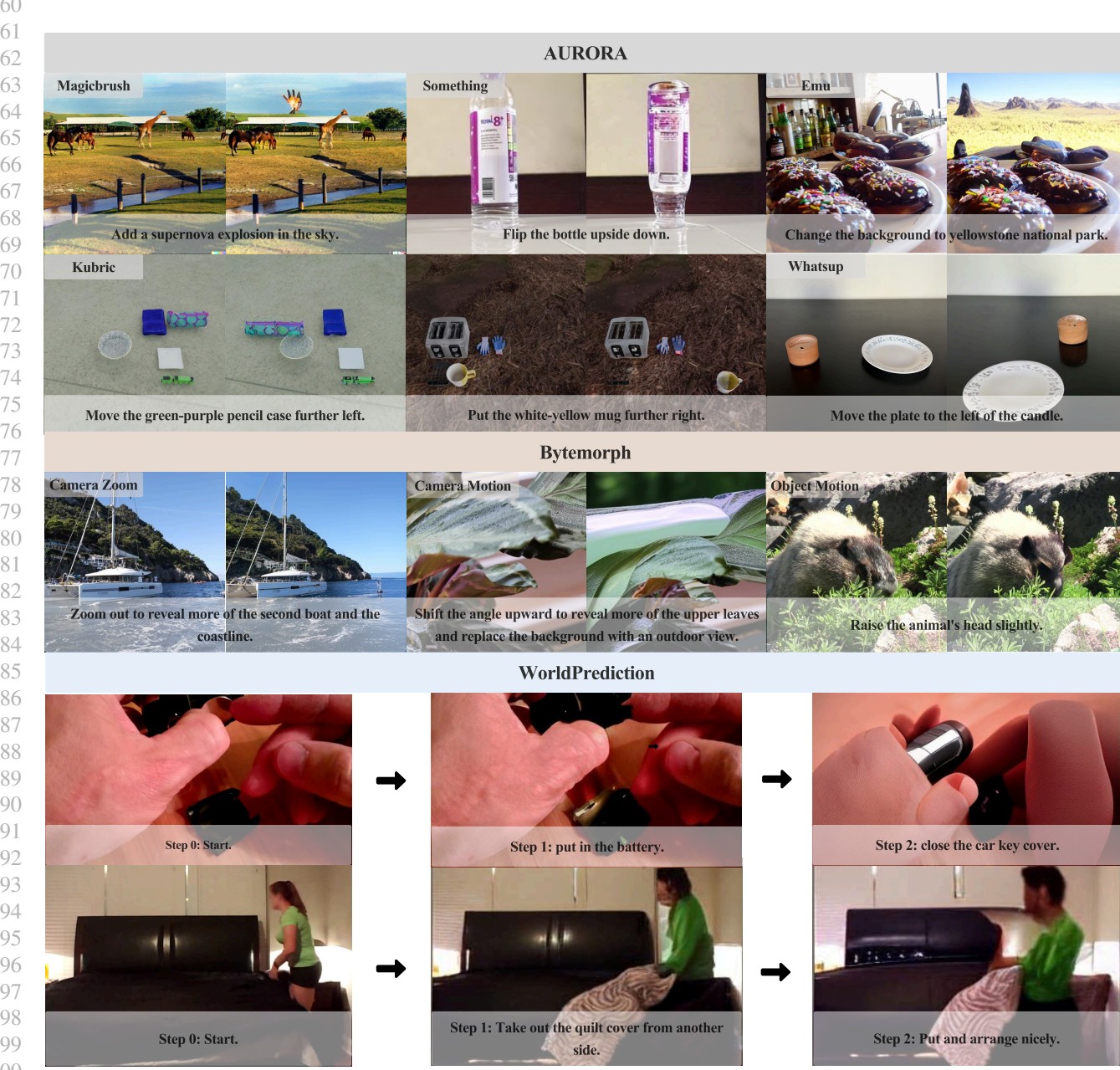

*Figure 7.* Qualitative examples produced by SWIRL for AURORA-BENCH, BYTEMORPH and WORLDPREDICTIONBENCH. We present the sample for each subset in these benchmarks.

