# OpenReview forum: "Self-Improving World Modelling with Latent Actions"
_ICML.cc/2026/Conference — Submitted to ICML 2026_

### Official Review · Reviewer_BQ72 · 2026-03-09

**Soundness:** 3
**Presentation:** 3
**Significance:** 3
**Originality:** 3
**Overall Recommendation:** 5
**Confidence:** 4

**Summary:**

This paper proposed a self-improvement framework that enhances world modeling capability for the LLMs and VLMs, named SWIRL. By treating actions as a latent variable, the framework optimizes Forward World Modelling (FWM) and Inverse Dynamics Modelling (IDM) alternately. To achieve self-improvement, SWIRL applies GRPO to optimize both the FWM and IDM. During the optimization of one model, the log probability produced by the other model is used as the reward signal, enabling learning from state-only sequences. Experiments on six benchmarks covering visual dynamics and text-based environments demonstrate significant improvements of SWIRL over the SFT baseline.

**Compliance With Llm Reviewing Policy:**

Affirmed.

**Final Justification:**

The rebuttal addresses most of my concerns. I appreciate the detailed response and keep my positive score.

**Key Questions For Authors:**

Please see weaknesses for details.

**Limitations:**

Yes

**Strengths And Weaknesses:**

Strengths:
1. The paper is well written and easy to follow. And the proposed self-improvement framework can be inspiring to the LLMs/VLMs research.
2. The theoretical framework is elegant and well-grounded. Mapping the alternating optimization of FWM and IDM to Variational Information Maximisation and ELBO Maximisation provides a principled foundation for the proposed self-improvement loop. The theorem derivations are logically structured and convincing.
3. The experimental coverage is broad, with thorough and comprehensive analyses. Evaluation across both VLM settings and LLM environments, including physical, web, and tool useing highlighting its generality.

Weaknesses and Questions:
1. The paper lacks direct comparisons with closely related prior approaches. Providing a more thorough comparison would help better clarify and strengthen the novelty of the proposed method.
2. Is it possible to reverse the order of Phase 1 and Phase 2? If so, how would the learning objectives or process change? If not, please explain this design.
3. For Phase 1, how does the high mutual information directly advance world modeling? Also, since the GRPO method sometimes exhibits entropy explosion or collapse, did you observe such behavior in your experiments? If so, does it affect the validity of Theorem 3.1?

---

> ### Author Rebuttal · Authors · 2026-03-31
>
> We thank the reviewer for the insightful reviews.
>
> ---
>
> **1. The paper lacks direct comparisons with closely related prior approaches. Providing a more thorough comparison would help better clarify and strengthen the novelty of the proposed method.**
>
> We would like to gently direct the reviewer to Table 1 and Section 4.1, where we explicitly evaluate SWIRL against the closest existing prior works: the Test-Time Verification and Bootstrap methods recently proposed in [1].
>
> In Test-Time Verification, the FWM generates multiple rollouts and is verified by the IDM to select the highest-probability sample, which motivates us to directly use IDM to reward FWM in the RL setup. Instead, the Boostrapping strategy learns an FWM from unlabelled videos relying on static pseudo-labelling from the IDM, followed by Supervised Fine-Tuning; however, visual transitions are inherently ambiguous (a single video can be described by many valid action texts). SFT forces strict, token-level imitation, which often leads to overfitting. SWIRL, instead, treats the action as a latent variable and uses RL (GRPO) to optimise for identifiability. We reward the Forward World Model (FWM) for generating any plausible future that the IDM can correctly infer the action from, avoiding the brittle exact-match constraint of SFT.
>
> As shown in our results, SWIRL significantly outperforms these baselines (e.g., our Iterative SWIRL achieves 5.06 vs. Bootstrap's 4.11 on Aurora-Bench).
>
> ---
>
> **2. Is it possible to reverse the order of Phase 1 and Phase 2? If so, how would the learning objectives or process change? If not, please explain this design.**
>
> Our design originates from the observation that IDM is much easier to learn than FDM in SFT as reported in the BoostrapWM [1] and ENACT [2] paper. Therefore, optimising the FWM with IDP first would provide a more stable learning process as the first stage.
>
> ---
>
> **3. For Phase 1, how does the high mutual information directly advance world modeling? Also, since the GRPO method sometimes exhibits entropy explosion or collapse, did you observe such behavior in your experiments? If so, does it affect the validity of Theorem 3.1?**
>
> We thank the reviewer for the very insightful comment. We actively monitored for entropy collapse (where the policy degenerates to outputting a single "safe" transition) and explosion (where outputs become unconstrained noise).
>
> As detailed in Appendix D and visualised in Figure 6, we empirically did not observe severe entropy collapse or explosion. Action uniqueness remained consistently high (>94%) across all iterations, and the semantic complexity (average token length) remained stable.
>
> In addition to the KL-divergence constraint in GRPO acting as an anchor to the prior reference policy, the reciprocal nature of SWIRL prevents the FWM from collapsing into a single "hackable" visual state, because the IDM is iteratively updated on the true data distribution (Phase II) to become a more rigorous critic.
>
> Regarding the impact on the validity of Theorem 3.1. Even if an RL run were to experience entropy collapse in practice, it would not invalidate the mathematical truth of Theorem 3.1. Theorem 3.1 establishes the theoretical property of the objective function, proving that maximising our specific reward formulation corresponds mathematically to maximising a lower bound on the Conditional Mutual Information.
>
> ---
>
> [1] Boostrapping visual dynamics in unified vision language model. Arxiv 2025.
>
> [2] ENACT: Evaluating Embodied Cognition with World Modeling of Egocentric Interaction. ICLR 2026.

---

> > ### Author Rebuttal · Reviewer_BQ72 · 2026-04-04
> >
> > The rebuttal addresses most of my concerns. I appreciate the detailed response and will keep my positive score.

---

### Official Review · Reviewer_wTSB · 2026-03-10

**Soundness:** 2
**Presentation:** 2
**Significance:** 3
**Originality:** 3
**Overall Recommendation:** 4
**Confidence:** 4

**Summary:**

The paper introduces a SWIRL framework, aimed at improving the world modelling capabilities of the foundation models like LLMs and VLMs, specifically from unlabelled action-less trajectories. The main contribution of the paper is the algorithm for learning both the Forward World Model (FWM) and Inverse Dynamics Model (IDM) by iteratively switching between training one or another. Each is trained using the GRPO algorithm in different reformulations. FWM is trained by sampling the actions from the frozen IDM, sampling the 1-step rollouts given the action from FWM, and taking the reward as logprob of the action given the state and the predicted rollout. IDM is trained similarly, but the reward is now logprob of the real next state given the sampled action.

**Compliance With Llm Reviewing Policy:**

Affirmed.

**Final Justification:**

Most of my concerns about the amount of compute and uncertainty estimates were addressed. However, the papers' ablations (section 4.5) need to be reconsidered, since the rebuttal revealed that the previous takeaways are based on statistically insignificant differences in scores.

**Key Questions For Authors:**

- Please, provide the uncertainty estimates (e.g. std with a few seeds) for the metrics in the paper to support the claims.
- The iterative two-model scheme using GRPO can be highly sample-inefficient. How much compute was spend on training on each phase relative to the SFT baseline?
- I also noted that a smaller and older baseline OmniGen (2024, 3.8B params) outperforms the method on Aurora-Bench and WorldPredictionBench. Can you explain that difference?

**Limitations:**

yes

**Strengths And Weaknesses:**

- The method is the combination of iterative learning of the latent dynamics model and the prediction model. Both of these were previously studied, but combining them in one algorithm is novel.
 - The paper builds heavily on the setup of Qiu et al. 2025, but the setup lacks an explanation for the inexperienced reader. What are the latent actions in each case? Are they arbitrary text, or specific tokens? The paper would benefit from the improved setup explanation by trading the space with the specific details of benchmarks and baselines, which could be moved to the appendix.
- The main weakness of the paper is the soundness of the experimental results. The tables 1-4 show blue cells as if the method outperforms the baseline, whereas the values could be the same (e.g., 0.92 vs. 0.92 for CLIP score on MagicBrush, 57.23 vs. 57.37 on Camera Zoom, etc.). This also transitions into the unsupported claims, like “shared weights introduce instability during training”, supported by a difference of 6.52 vs. 6.57, which is a <1% difference in relative terms. Considering the inherent noiseness of the LLM-as-a-judge validation and the baseline on multiple benchmarks (e.g., ScienceWorld, Mind2Web), the paper really needs the uncertainty estimates.

---

> ### Author Rebuttal · Authors · 2026-03-31
>
> We thank the reviewer for the insightful reviews.
>
> ---
>
> **1. The setup of this paper lacks an explanation. What are the latent actions? Are they arbitrary text, or specific tokens?**
>
> Latent actions are arbitrary text token sequences (actions in language form), rather than a predefined set of specific, discrete action tokens. We agree that we will expand the discussion of test-time verification and boostrapping FWM with IDM-labelled synthesised trajectories in the Baselines section. We will also expand the explanation for Metrics: 1) CLIP score: we use the CLIP-image encoder to encode the predicted and reference images, and compute the cosine similarity between their embeddings. 2) DiscEdit: this metric serves as an auxiliary indicator to reduce the model's bias towards copying source observation as the target, which inflates CLIP scores; instead, DiscEdit credits the predicted image when it is close to the target image, while far away from the source image.
>
> ---
>
> **2. The paper really needs the uncertainty estimates.**
>
> We are happy to report the mean and standard deviation of three runs of experiments on AURORA, ByteMorph and WorldPredictionBench to address the reviewers' concern:
>
> | AURORA | MB | AG | ST | WU | KU | Avg. |
> |-|-|-|-|-|-|-|
> | LIQUID-SFT | 6.1 ± 0.07 | 3.20 ± 0.18 | 3.57 ± 0.20 | 4.53 ± 0.06 | 6.04 ± 0.31 | 4.68 ± 0.11 |
> | SWIRL (ITERATIVE) | 6.47 ± 0.23 | 3.48 ± 0.13 | 3.81 ± 0.28 | 4.41 ± 0.12 | 6.95 ± 0.15 | 5.02 ± 0.10 |
> | SWIRL (ITER.+SHARE) | 6.40 ± 0.34 | 3.23 ± 0.18 | 3.82 ± 0.31 | 4.56 ± 0.03 | 7.09 ± 0.25 | 5.02 ± 0.19 |
> | SWIRL (IDM->FWM) | 6.33 ± 0.29 | 3.40 ± 0.17 | 3.95 ± 0.07 | 4.29 ± 0.20 | 6.68 ± 0.25 | 4.93 ± 0.15 |
>
> | ByteMorph | Camera Zoom | Camera Motion | Object Motion | Human Motion | Interaction | Overall |
> |-|-|-|-|-|-|-|
> | LIQUID-SFT | 50.83 ± 3.86 | 50.28 ± 1.81  | 42.13 ± 0.58  | 35.37 ± 0.24 | 39.74 ± 1.00 | 40.86 ± 0.50 |
> | SWIRL (ITERATIVE) | 54.83 ± 5.02 | 56.53 ± 2.44  | 57.59 ± 0.96  | 48.15 ± 1.30 | 52.85 ± 2.47 | 52.93 ± 0.69 |
> | SWIRL (ITER.+SHARE) | 53.47 ± 3.92 | 54.47 ± 2.00  | 61.03 ± 3.53  | 51.65 ± 1.53 | 53.13 ± 0.90 | 54.48 ± 1.03 |
> | SWIRL (IDM → FWM) | 54.96 ± 1.81 | 53.33 ± 3.60  | 59.38 ± 3.26  | 47.19 ± 1.07 | 54.87 ± 1.24 | 53.21 ± 1.36 |
>
> | WorldPredictionBench  | Avg. | T=1 | T=2 | T=3 | T=4 | T=5 | T=6 |
> |-|-|-|-|-|-|-|-|
> | LIQUID-SFT | 1.79 ± 0.04 | 3.12 ± 0.02 | 1.99 ± 0.08 | 1.41 ± 0.02 | 1.19 ± 0.03 | 1.09 ± 0.07 | 0.97 ± 0.05 |
> | SWIRL (ITERATIVE) | 1.88 ± 0.04 | 3.11 ± 0.10 | 2.03 ± 0.03 | 1.57 ± 0.05 | 1.36 ± 0.04 | 1.21 ± 0.04 | 1.10 ± 0.03 |
> | SWIRL (ITER.+SHARE)  | 1.86 ± 0.03 | 2.99 ± 0.06 | 2.04 ± 0.02 | 1.55 ± 0.03 | 1.34 ± 0.08 | 1.24 ± 0.02 | 1.10 ± 0.01 |
> | SWIRL (IDM → FWM) | 1.96 ± 0.04 | 3.23 ± 0.04 | 2.15 ± 0.05 | 1.57 ± 0.04 | 1.37 ± 0.03 | 1.20 ± 0.04 | 1.10 ± 0.05 |
>
> The performance gaps between the SWIRL variants and the LIQUID-SFT baseline exceed the standard deviations across macro-averages. For instance, on ByteMorph, the Overall score jumps from 40.86 (±0.50) to 52.93 (±0.69) using SWIRL. This demonstrates that our performance gains are significant.
>
> [1] Learning Action and Reasoning-Centric Image Editing from Videos and Simulations. NeurIPS 2024.
>
> ---
>
> **3. How much compute was spend on training on each phase relative to the SFT baseline?**
>
> On NVIDIA H200 GPUs, SFT required 211.2 GPU hours. For SWIRL with a rollout size of G=16, the FWM training phase required 249.6 GPU hours, and the IDM training phase required 44.96 GPU hours. SWIRL optimisation requires approximately ~1.4× the compute of standard SFT. Given the known computational demands of RL generation rollouts, we consider this a highly practical and tractable overhead.
>
> We respectfully emphasise that while GRPO spends more compute per sample (due to rollouts), it is actually more sample-efficient in raw data required to learn FWM. As demonstrated in our apples-to-apples ablation in Figure 3, SWIRL achieves higher performance than SFT when restricted to the same number of unlabelled training samples. While SFT quickly plateaus and overfits to the single pseudo-label by the IDM, SWIRL uses the rollout space to discover valid dynamic pathways.
>
> ---
>
> **4. Can you explain why OmniGen (2024, 3.8B params) outperforms the method on Aurora-Bench and WorldPredictionBench?**
>
> OmniGen relies on a diffusion backbone, which excels at high-fidelity generation and precise editing. In contrast, Liquid-7B is a purely Autoregressive (AR) multimodal model that operates on discrete visual tokens, which naturally struggle to match the granular spatial control and visual consistency of dedicated diffusion models on zero-shot or complex editing benchmarks like AURORA. Our choice to use a purely AR model (Liquid) was a pragmatic decision: it allows for the seamless, off-the-shelf application of discrete-token RL (GRPO) and its infrastructure without the extreme memory and computational complexities currently required to run RL directly through diffusion generation steps.

---

> > ### Author Rebuttal · Reviewer_wTSB · 2026-04-03
> >
> > Thank you for the clarifications and additional experiments.
> >
> > The method clearly outperforms the Liquid-SFT baseline.  However, I note that the difference in performance of the three versions (ITERATIVE, ITER.+SHARE, IDM → FWM) is negligible on 2 of the 3 benchmarks, which means that the paper ablations (section 4.5), specifically regarding weight sharing and the number of phases needed for convergence, should be reconsidered and rewritten.
> >
> > Apart from that, the rest of my concerns are addressed. I am raising the score to 4.

---

### Official Review · Reviewer_J7TF · 2026-03-12

**Soundness:** 3
**Presentation:** 3
**Significance:** 4
**Originality:** 4
**Overall Recommendation:** 5
**Confidence:** 3

**Summary:**

The paper introduces SWIRL, a framework designed to train world models within LLMs and VLMs using unlabelled state-transition sequences. SWIRL treats the intermediate actions as latent variables and establishes a training cycle between a forward an inverse dynamics models. Phase I updates the FWM using GRPO to generate next states that maximise the conditional mutual information with the latent actions, using the frozen IDM as a reward signal. Phase II updates the IDM to maximize the ELBO of the observed transitions, using the frozen FWM as the reward.

**Compliance With Llm Reviewing Policy:**

Affirmed.

**Final Justification:**

SWIRL is a well-grounded and innovative method that formulates FWM and IDM as an alternating policy-reward duo via GRPO to discover latent actions. The paper is clearly written; the alternating updates are convincingly tied to variational inference. My main concern was the reliance on the SFT warmup phase and its implications for the "unlabeled" claim. The rebuttal addressed this by clarifying that the warm-up was needed only to adapt models like Liquid 7B to a new input-output format, and that newer natively multimodal VLMs (e.g., Emu3.5-30B) remove this requirement. The authors also gave thoughtful answers on initialisation sensitivity, prior PEFT exploration, and a promising direction for multi-step transition rewards. I maintain my positive assessment of the paper and recommend it for acceptance.

**Key Questions For Authors:**

1. Appendix C notes that omitting PICO-BANANA-400K during the SFT warmup degrades performance. How sensitive is SWIRL to the diversity and volume of this initial data? I'm curious if the framework could recover from a much weaker initialisation if given more RL iterations.
2. In the shared weights (ϕ=θ) ablation, you observed instability and a performance drop at iteration 3 due to gradient interference between generative and understanding heads. Have you explored using PEFT methods to mitigate this?
3. For multi-turn predictions, compounding covariate shift still degrades performance over time. Could SWIRL be adapted to incorporate multi-step transition rewards?

**Limitations:**

yes

**Strengths And Weaknesses:**

### Strengths

- The theoretical foundation of the paper is sound. The authors prove that their alternating GRPO updates correspond to variational information maximisation for the FWM and ELBO maximisation for the IDM.
- The paper is logically structured and clearly written.
- I think this work unlocks a highly practical pathway for training better foundation models.
- While reciprocal reward models and self-play are active research areas, formulating the FWM and IDM as an alternating policy-reward duo using GRPO to discover latent actions is quite innovative.

### Weaknesses
- The method relies on the SFT warm-up phase to initialise the policy. The authors acknowledge that without this, the model outputs invalid actions/states, making RL exploration impossible. I think this limits the claim that the model learns purely from unlabelled data, as the quality of this prior affects the success of the self-improvement loop.

---

> ### Author Rebuttal · Authors · 2026-03-31
>
> We sincerely thank the reviewer for the constructive feedback and insightful comments. We address your specific points below:
>
> ---
> **1. The method relies on the SFT warm-up phase to initialise the policy. The authors acknowledge that without this, the model outputs invalid actions/states, making RL exploration impossible.**
>
> We completely agree that a suitable prior is necessary to make RL-based exploration meaningful. At the time of submission, the available VLMs compatible with our compute limits (such as Liquid 7B) natively supported only text-to-image generation. Hence, the SFT phase was strictly employed as a warm-up stage to equip the model with a new input-output format, namely image+text to image. More recent and larger auto-regressive VLMs, such as Emu3.5-30B, already natively support this format, hence there is no need for such a warm-up stage anymore. We will include a clarification on which types of models need this SFT warm-up stage in the camera-ready version.
>
> ---
> **2. Appendix C notes that omitting PICO-BANANA-400K during the SFT warmup degrades performance. How sensitive is SWIRL to the diversity and volume of this initial data?**
>
> For GRPO to succeed, the FWM must occasionally generate an image that is physically coherent enough for the IDM to recognise the latent action and provide a meaningful reward.
> If the initialisation is too weak (e.g., omitting PICO-BANANA-400K makes it harder to adapt to the new image+text to image format), the FWM's rollouts are largely non-meaningful images. Consequently, the IDM's reward signal becomes sparse or noisy, leading to a "cold start" failure where more RL iterations simply cause the policy to collapse rather than recover.
>
> ---
> **3. In the shared weights (ϕ=θ) ablation, you observed instability and a performance drop at iteration 3 due to gradient interference between generative and understanding heads. Have you explored using PEFT methods to mitigate this?**
>
> We ran an early experiment with LoRA; however, we found that PEFT leads to very poor generalisation in the out-of-distribution splits in AURORA. Specifically, we fine-tuned a model with LoRA on (magicbrush, action genome, something-something, and kubric) and observed very poor generalisation on unseen datasets such as Whatsup (LoRA at 0.84 compared to full-model fine-tuning at 2.88 according to GPT40-as-judge score) in AURORA. This initial exploration led us to stick with full-model fine-tuning. Since LoRA fails in "simpler" SFT settings, it would not make sense to use it in SWIRL with the "in-the-wild" videos from OpenVID.
>
> ---
> **4. For multi-turn predictions, compounding covariate shift still degrades performance over time. Could SWIRL be adapted to incorporate multi-step transition rewards?**
>
> We strongly agree with this suggestion. Compounding covariate shift is indeed the primary bottleneck for autoregressive multi-turn prediction. Extending SWIRL to incorporate multi-step transition rewards is a highly promising formulation that we are actively exploring for future work.
>
> Overall, an important concept outlined by the manuscript is the reciprocal FWM-IDM loop, which inherently supports multi-step scaling. In a multi-step adaptation, the FWM could generate a rollout trajectory $x_1, \dots, x_T$ given a sequence of latent actions $z_{1:T}$. The IDM could then be prompted to evaluate the global temporal consistency, either by recursively predicting the action sequence from the start and end states $(x_0, x_T)$, or by providing a sequence-level reward.

---

> > ### Author Rebuttal · Reviewer_J7TF · 2026-04-01
> >
> > Thank you for your response. I am satisfied with the answers and have no further questions at this point.

---

### Official Review · Reviewer_SJyS · 2026-03-13

**Soundness:** 3
**Presentation:** 4
**Significance:** 3
**Originality:** 3
**Overall Recommendation:** 4
**Confidence:** 3

**Summary:**

The paper introduce SWIRL, a unified framework for enabling VLMs and LLMs to intrinsically model future states. By interpreting actions as latent variables and alternately optimising forward world modelling and inverse dynamics modelling objectives with GRPO, the work induces self-improving world modelling purely from unlabelled data. They demonstrate theoretical and empirical analysis on several visual and textual environments.

**Compliance With Llm Reviewing Policy:**

Affirmed.

**Key Questions For Authors:**

Refer Weakness section

**Limitations:**

Yes

**Strengths And Weaknesses:**

Strengths
- The authors propose a novel self-improving framework for world modelling in LLMs and VLMs, reciprocally reinforcing FWM and IDM without annotated labels.
- Paper provides a rigorous theoretical proof that SWIRL corresponds to alternating between maximising Variational Mutual Information and Evidence Lower Bound.
- Empirical evidence on six benchmarks across visual, textual, web, tool calling environments demonstrates effectiveness of SWIRL

Weaknesses
- For evaluation, GPT-4o-as-a-judge is used extensively in table 2 and 3. However, gpt-40 scores are known for its biases (position, verbosity bias). It will be nice to mention its biases as a limitation. Also, what do you think about evaluation with a alternate metrics (CLIP/ Dino scores) as in Table 1. Or, perceptual quality metrics such as SSIM, LPIPS, FVD for videos, particularly for World PredictionBench.

- In L92, the claim of "learning from unlabelled state-only sequences" appears overstated, since all experiments include an SFT warm-up phase prior to RL training, as described in Section 5.

- In abstract, the paper claims to 'provide theoretical learnability guarantees for both updates'. The theorems do not discuss the tightness of these bounds. e.g. Theorem 3.2 shows that the GRPO objective equals a beta-ELBO, but the ELBO is a lower bound on the marginal likelihood, and coordinate ascent on the ELBO is not guaranteed to converge to the global maximum of the marginal likelihood, particularly in non-convex settings

Minor
- The abstract reports relative improvements, but it does not specify the reference point—that is, whether these gains are measured relative to Liquid-SFT or to the strongest prior baseline?
- typo 'FDM' in Fig 1.

---

> ### Author Rebuttal · Authors · 2026-03-31
>
> **1. GPT4o's biases and add additional metrics for ByteMorph and WorldPredictionBench.**
>
> We use GPT4o score because a reference-free metric is more reliable for world modelling, given its inherent ambiguity: multiple target observations are valid for a source observation and action. However, we agree to include additional metrics: the editing success and CLIP as in the AURORA, as well as SSIM and LPIPS as multi-frame evaluation for WorldPredictionBench.
>
> **ByteMorph**:
>
> | Model | GPTScore | Edit Success | CLIP |
> |-|-|-|-|
> | LIQUID-SFT | 40.86 | 0.09 | 0.88 |
> | SWIRL (ITERATIVE) | 52.93 | 0.16 | 0.87 |
> | SWIRL (ITER.+SHARE) | 54.48 | 0.18 | 0.86 |
> | SWIRL (IDM->FWM) | 53.21| 0.15 | 0.87 |
>
> We find that SFT and SWIRL variants have a similar CLIP score; however, crucially, the editing success score for SWIRL is much higher, indicating that SFT tends to copy the source observation directly as the target, resulting in an inflated CLIP, as observed in [1].
>
> **WorldPredictionBench**:
>
> | SSIM (↑)| COIN | CROSS | EgoExo | EPIC | IKEA | Avg. | T=1 | T=2 | T=3 | T=4 | T=5 | T=6 |
> |-|-|-|-|-|-|-|-|-|-|-|-|-|
> | LIQUID-SFT | 0.4505 | 0.3952 | 0.3706 | 0.3230 | 0.4609 | 0.4001 | 0.4571 | 0.4234 | 0.4148 | 0.4058 | 0.3953 | 0.3955 |
> | SWIRL (ITERATIVE) | 0.4471 | 0.3927 | 0.3701 | 0.3301 | 0.4613 | 0.4003 | 0.4521 | 0.4242 | 0.4153 | 0.4078 | 0.3985 | 0.3964 |
> | SWIRL (ITER.+SHARE) | 0.4465 | 0.3952 | 0.3697 | 0.3320 | 0.4618 | 0.4010 | 0.4517 | 0.4244 | 0.4163 | 0.4077 | 0.3998 | 0.3974 |
> | SWIRL (IDM → FWM) | 0.4457 | 0.3954 | 0.3705 | 0.3259 | 0.4634 | 0.4002 | 0.4530 | 0.4230 | 0.4149 | 0.4076 | 0.3997 | 0.3965 |
>
> | LPIPS (↓) | COIN | CROSS | EgoExo | EPIC | IKEA | Avg. | T=1 | T=2 | T=3  | T=4 | T=5 | T=6 |
> |-|-|-|-|-|-|-|-|-|-|-|-|-|
> | LIQUID-SFT | 0.7042 | 0.6935 | 0.6245 | 0.7021 | 0.6491 | 0.6747 | 0.5753 | 0.6748 | 0.7007 | 0.6996 | 0.6924 | 0.7015 |
> | SWIRL (ITERATIVE) | 0.7041 | 0.6950 | 0.6147 | 0.6958 | 0.6351 | 0.6689 | 0.5884 | 0.6700 | 0.6927 | 0.6864 | 0.6719 | 0.6805 |
> | SWIRL (ITER.+SHARE) | 0.7037 | 0.6921 | 0.6129 | 0.6994 | 0.6358 | 0.6688 | 0.5891 | 0.6708 | 0.6916 | 0.6855 | 0.6734 | 0.6820 |
> | SWIRL (IDM → FWM) | 0.6989 | 0.6877 | 0.6133 | 0.6985 | 0.6372 | 0.6671 | 0.5849 | 0.6681 | 0.6905 | 0.6858 | 0.6750 | 0.6848 |
>
> All SWIRL variants outperform LIQUID-SFT slightly in terms of this metric (having a lower avg), with SWIRL (IDM→FWM) best at 0.6671 vs 0.6747. The trend across turns is quite consistent with GPT4o-as-judge, where the gap between SWIRL and Liquid-SFT widens as the turns increase. This indicates that our method is more advantageous (e.g., SWIRL 0.3974 vs SFT 0.3955 on SSIM and 0.682 vs 0.7015 on LPIPS in T=6) for world modelling with a longer horizon.
>
> [1] Learning Action and Reasoning-Centric Image Editing from Videos and Simulations. NeurIPS 2024.
>
> ---
> **2. All experiments include a SFT warm-up phase prior to RL training.**
>
> To clarify, our base models (e.g., Liquid-7B) lack the native ability to generate images conditioning on image + text. As detailed in Section 4.1 and our Sanity Check in Appendix C, the SFT phase is strictly an initialisation step to format the model's outputs. Importantly, the SFT data consists only of general image editing (e.g., add/replace/remove). The actual world modelling capabilities (such as object motion, human interaction, and physics) are learned entirely from the unlabelled state-only sequences (e.g., the 1M unlabelled videos from VIDGen-1M) during the SWIRL phase. As shown in Table 2, the SFT model alone achieves only 43.38 on ByteMorph, whereas SWIRL elevates this to 53.77 (+26.4% relative gain) using purely unlabelled transitions from OpenVID-1M. We will make this distinction more explicit in the revised manuscript.
>
> ---
> **3. The theorems do not discuss the tightness of these bounds.**
>
> In our revision, we will replace "theoretical learnability guarantees" in the abstract and introduction with "theoretical justifications" and "formal connections to variational bounds." We will also add a paragraph in Section 3.3 explicitly discussing the limitations of coordinate ascent in non-convex landscapes and noting that the tightness of our ELBO depends on the capacity of the IDM acting as the variational posterior.
>
> Our primary goal is to provide a principled foundation for SWIRL. Theorems 3.1 and 3.2 are significant because they formally prove that applying GRPO with our log-likelihood reward definitions is not an ad-hoc heuristic, but exactly corresponds to performing coordinate ascent on well-established variational objectives (a lower bound on CMI for FWM, and a $\beta$-ELBO for IDM). While this does not guarantee convergence to the global marginal likelihood maximum, it guarantees that our iterative updates are optimising a valid proxy that encourages data fidelity and identifiability.
>
> ---
> **4. Presentation issues**
>
> Thanks. Here we are comparing with the Liquid-SFT. We will mention it in the abstract, and fix the Fig1 typo.

---

> > ### Author Rebuttal · Reviewer_SJyS · 2026-04-04
> >
> > Thanks authors for addressing the concerns.

---

### Decision · Program_Chairs · 2026-04-30

**Decision:**

Reject

**Comment:**

The paper presents an interesting post-training framework for improving world-modeling behavior in LLMs and VLMs, and the reviewers generally found the method technically reasonable. The rebuttal also addressed several concerns about evaluation, initialization, and theoretical claim calibration.

That said, I am not convinced the current submission is strong enough for acceptance. Consistent with the post-rebuttal reviewer discussion, some of the more fine-grained empirical conclusions remain less convincing than the paper suggests, particularly the takeaways drawn from the SWIRL ablations. In addition, I view the contribution as a post-training systems framework rather than a fundamentally new world-modeling paradigm, and the broader positioning is weakened by limited discussion of prior latent-action world model literature (Genie, Adaworld, etc., see references below).

Overall, I view this as a promising direction with a meaningful systems contribution, but not yet a sufficiently strong case for acceptance in its current form.

[1] Bruce, Jake, et al. "Genie: Generative interactive environments." Forty-first International Conference on Machine Learning. 2024.
[2] Gao, Shenyuan, et al. "Adaworld: Learning adaptable world models with latent actions." arXiv preprint arXiv:2503.18938 (2025).
[3] Wang, Yucen, et al. "Co-Evolving Latent Action World Models." arXiv preprint arXiv:2510.26433 (2025).
[4] Bi, Hongzhe, et al. "Motus: A unified latent action world model." arXiv preprint arXiv:2512.13030 (2025).